# Nutritional Composition and Microbial Communities of Two Non-alcoholic Traditional Fermented Beverages from Zambia: A Study of *Mabisi* and *Munkoyo*

**DOI:** 10.3390/nu12061628

**Published:** 2020-06-01

**Authors:** Justin Chileshe, Joost van den Heuvel, Ray Handema, Bas J Zwaan, Elise F. Talsma, Sijmen Schoustra

**Affiliations:** 1Laboratory of Genetics, Wageningen University and Research, P.O. Box 16, 6700AA Wageningen, The Netherlands; chileshej@tdrc.org.zm (J.C.); joost.vandenheuvel@wur.nl (J.v.d.H.); bas.zwaan@wur.nl (B.J.Z.); 2Tropical Diseases Research Centre, Department of Biomedical Sciences, P.O. Box 71769, Ndola 10101, Zambia; handemar@tdrc.org.zm; 3Division of Human Nutrition and Health, Wageningen University, P.O. Box 57, 6700AB Wageningen, The Netherlands; elise.talsma@wur.nl; 4Department of Food Science and Nutrition, School of Agricultural Sciences, University of Zambia, P.O. Box 32379, Lusaka 10101, Zambia

**Keywords:** *Mabisi*, *Munkoyo*, traditional fermentation, Zambia, lactic acid bacteria, nutritional composition, nutrient value, proximate analysis

## Abstract

Traditional fermented foods and beverages are common in many countries, including Zambia. While the general (nutritional) benefits of fermented foods are widely recognised, the nutritional composition of most traditional fermented foods is unknown. Furthermore, fermentation is known to add nutritional value to raw materials, mainly by adding B-vitamins and removing anti-nutritional factors. In the case of traditional fermentation, the composition of microbial communities responsible for fermentation varies from producer to producer and this may also be true for the nutritional composition. Here, we characterized the nutrient profile and microbial community composition of two traditional fermented foods: milk-based *Mabisi* and cereal-based *Munkoyo.* We found that the two products are different with respect to their nutritional parameters and their microbial compositions. *Mabisi* was found to have higher nutritional values for crude protein, fat, and carbohydrates than *Munkoyo*. The microbial community composition was also different for the two products, while both communities were dominated by lactic acid bacteria. Our analyses showed that variations in nutritional composition, defined as the amount of consumption that would contribute to the estimated average requirement (EAR), might be explained by variations in microbial community composition. Consumption of *Mabisi* appeared to contribute more than *Munkoyo* to the EAR and its inclusion in food-based recommendations is warranted. Our results show the potential of traditional fermented foods such as *Mabisi* and *Munkoyo* to add value to current diets and suggests that variations in microbial composition between specific product samples can result in variations in nutritional composition.

## 1. Introduction

In many countries, locally processed traditional foods exist and these contribute to the diets of their consumers. Yet, for many of these products, the methods of preparation are not uniform and documented, their functional properties such as product composition, organoleptic characteristics and shelf life are unknown and the way these affect their nutritional composition has not been assessed. As a result, these local traditional foods are often not included in food-based dietary guidelines nor in estimations of how they can contribute to local food and nutrition security. 

Fermented foods and beverages that are produced using traditional fermentation processes are of special interest, since these foods are locally available and are a part of tradition. Like in other fermented foods, fermentation adds value to the raw materials used, resulting in a product with a prolonged shelf-life and stability and an increased sensory, and monetary value [1,2]. Furthermore, the activity of micro-organisms is known to add nutritional value to raw materials, for instance by the production of B-vitamins and the removal of anti-nutritional factors such as phytate. Removal of phytate increases the bioavailability of various micronutrients [3,4]. In milk-based fermented foods, the anti-nutritional factor lactose is converted into lactic acid during fermentation. Removing lactose has been linked to health benefits by reducing abdominal pain and diarrhoea in people with lactose intolerance [5,6,7]. As a result, the final nutritional and sensory properties of fermented products depend on their diverse microbial community. In turn, the composition of the microbial community to a large extent depends on the raw ingredients of each geographical region and traditional processing procedures [8,9]. Apart from increased nutritional contents compared to raw materials, fermented foods possess beneficial effects on human health, for example, through the modification of gut microbiota leading to a better immune response and the lowering of a person’s risk of hypertension, diabetes, and high cholesterol [10]; the prevention and treatment of inflammatory bowel disease (IBD) [11]; and anti-carcinogenic and hypo-cholesterolemic effects [12]. 

In Zambia, various traditional non-alcoholic fermented foods exist that are consumed by all age groups. Of these, *Mabisi* and *Munkoyo* are commonly found in rural areas. *Munkoyo* is also found in some urban areas. Although consumption of both products is frequent and the product is an important part of the local diet [13], surprisingly, the nutritional composition has not been characterized. *Mabisi* is produced by fermenting raw cow’s milk and *Munkoyo* is produced by fermenting maize porridge [13,14]. *Mabisi* is made by placing raw milk in a fermentation vessel and fermenting at ambient temperatures for 48 h, resulting in a mildly sour tasting product. Previous research has shown that variations in processing exist, which could lead to variations in product functionality in terms of microbial composition and sensory properties [13,14]. Processing most notably differs in the repeated additions (or not) of fresh milk, the level of shaking during the fermentation, and the levels of back-slopping (transfer of material from an old batch to a new batch, [14]). The traditional fermented food *Munkoyo* is made from maize flour that is mixed with water and boiled for several hours. After cooling, *Rhynchosia* roots are added to provide enzymes to degrade complex sugars and to provide a microbial inoculum for fermentation [15]. Fermentation can be done in a variety of vessels at ambient temperatures and takes around 48 h. Processing variations include the time allowed for cooking the maize porridge, the types of roots added, the fermentation vessel used, and the level of back-slopping [13]. 

For both traditional fermented food products, the microbial communities responsible for fermentation are dominated by four to ten species of lactic acid and acetic acid bacteria [13,14,16]. The exact composition varies between samples of the same product, and variations in processing, such as the containers used for processing, gives rise to further differentiation in microbial composition [13,14,17]. Since micro-organisms affect the nutrient composition and increase the nutritional value, variations in microbial composition may lead to variation in nutritional value of the final products. A previous study has compared microbial community structure of fermented microbes in the two types of traditional fermented foods *Mabisi* and *Munkoyo* [13]. While expecting a clear signature of raw materials used as a driver of the composition of the microbial community of fermenting microbes, these results were inconclusive due to uncontrolled factors such as geographical region, climatic conditions and processing variation. 

In the present study we analysed product samples of the traditional fermented foods *Mabisi* and *Munkoyo* that we collected from local producers in Zambia. We documented the nutritional composition of *Mabisi* and *Munkoyo* and their variations among products from different producers and profiled the microbial communities that are present at the end of fermentation. We expected to find variations in both nutritional composition and microbial community profiles between the different products and among samples taken of the same product. Finally, we assessed whether we could correlate variations in microbial communities to variations in nutritional content of the products. Our study thus provides unique data on the nutritional composition of two traditional fermented foods that could be part of the new food-based dietary guidelines currently in development for Zambia. 

## 2. Materials and Methods 

### 2.1. Study Design

This was a cross sectional study focusing on the nutritional composition and microbial community composition of the traditional fermented foods *Mabisi* and *Munkoyo.* Samples were collected from the Mkushi area in Zambia (location coordinates 13.1339° S, 27.8493° E). This site was selected because of the tradition of making *Mabisi* and *Munkoyo* that has been maintained by the collection of people who have migrated from other parts of Zambia to live among the locals (Swaka people), and because *Mabisi* and *Munkoyo* are locally produced in this area.

Samples were purchased from producers either at their homes or at the market where they were selling their products; 12 *Mabisi* and 13 *Munkoyo* samples were collected. Producers were selected on the basis of their location, their presence at the market in Mkushi and by their processing method for the production of *Mabisi* and *Munkoyo*. All *Mabisi* processors were selected to use the Tonga-type method of fermentation [14]. Tonga is a tribe traditionally located in the southern part of Zambia. This method is characterized by placing raw cow’s milk into a container to allow fermentation for 48 h at an ambient temperature in unshaken containers without the addition of fresh milk during fermentation nor the removal of whey that occurs due to whey separation. All *Munkoyo* processors were selected to use the Kitwe method of fermentation [16,17]. This method is characterized by mixing maize meal with water and cooking this for 30 min to gelatinize the starch. After cooling, *Rhynchosia* roots are added and the mixture is placed in a fermentation vessel to allow fermentation for 48 h at an ambient temperature. For both traditional fermented foods, the vessels used for fermentation may vary. We recorded the type of fermentation vessel (such as plastic bottles or buckets, metal cans or calabashes) that the processors had used as this may have had an impact on the microbial composition [18]. Samples were collected in two duplicate sterile 50 mL centrifuge tubes with screw caps and were immediately placed in a cool box with ice packs after which they were stored in the freezer, one tube at −20 °C (for nutritional analysis) and the other tube at −80 °C (for microbial analysis). Samples were analysed at the Tropical Diseases Research Centre (TDRC) in Zambia for pH, B-vitamin and mineral (calcium, iron and zinc) concentrations and at the University of Zambia, School of Agricultural Sciences, Department of Food Sciences and Nutrition for proximate content (protein, fibre, water, fat, energy and carbohydrates). Samples were transported to Wageningen University in the Netherlands for whole bacterial genomic DNA extraction and sample preparation for 16S rRNA amplicon sequencing.

This study was approved by the TDRC Ethics review committee (Ndola, Zambia) (Ethics number STC/2015/13). 

### 2.2. Measurement of Nutrient Value

Our study assessed the levels of the main components of the products (including dry matter, carbohydrates, fats, protein, energy, fibre and ash), B-vitamins (B1, B2, B3, B6 and B12), and minerals (calcium, iron and zinc) using customarily used methods of analysis. Furthermore, the energy content for each sample was calculated. We chose to measure levels of selected minerals and B-vitamins since these are the most relevant considering the raw materials used in the production of *Mabisi* (milk) and *Munkoyo* (maize). We calculated the contribution of one adult portion size of 183 g [19] to reach the Estimated Average Requirement (EAR) for each of the nutritional components. Iron bioavailability was estimated at 5% and low bioavailability for Zinc was applied. All EAR’s were based on WHO/FAO recommendations [20], however for protein no WHO/FAO EAR exist and therefore 80% of the Population Reference Intake of EFSA [21] was used for an average women of 60 kg. 

#### 2.2.1. Proximate Analysis

Different proximate parameters in *Munkoyo* and *Mabisi* were determined using the methods of the Association of Official Analytical Chemists (AOAC) [22]. Briefly, crude fat was measured in samples using Soxhlet apparatus with hexane as the solvent in the AOAC procedure. Protein was determined using nitrogen content by the micro-Kjeldahl method where nitrogen value obtained for each sample was converted to crude protein by multiplying it with the 6.25 factor for *Munkoyo* and the 6.38 factor for *Mabisi*. Moisture and dry matter content were determined by weighing 2 g of sample onto a crucible, heating it to dryness in an oven at 110 °C for 2 h and calculating the weight difference. Crude fibre was determined in each sample after the removal of fat using successive digestion with 1.25% sulphuric acid and 1.25% sodium hydroxide solutions. Carbohydrate content was determined by a difference calculation method as follows: % Total Carbohydrate = [100 − % (Protein + Fibre+ Ash + Fat + Moisture)]. All the proximate parameters were reported in AOAC, 2000 standard format as percentage.

#### 2.2.2. Atomic Absorption Spectrophotometry (AAS)

The contents of calcium, iron and zinc were determined by dry ashing samples at 450 °C in a muffle furnace following the procedures described earlier [23]. After dissolution of the resulting ash in hydrochloric acid (HCl), the metal element contents in the solutions were determined by flame atomic absorption spectrophotometry (AAS) (AAnalyst-400, Perkin-Elmer Corp., Norwalk, CT, USA). Standards were prepared from stock standard solutions (1000 ppm) of zinc, iron and calcium to make a calibration curve for each element. Analyses were performed in duplicate. 

#### 2.2.3. Quality Control (QC)

To monitor performance and reproducibility of the analytical procedures used, we included quality control samples for each batch of samples on a daily basis. For calcium and iron we used Cobas™ control samples while for zinc we used inhouse QC samples with values previously determined using the Seronorm™ Trace Elements Serum Level 1 and 2. The means, standard deviations and coefficient of variation (CV%) for duplicate samples were calculated to ensure that the values were within the acceptable limits (10%). 

#### 2.2.4. Analysis of B-Vitamins (B1, B2, B3, B6 and B12) by High Pressure Liquid Chromatography (HPLC)

A previous method based of HPLC UV analysis was used with a few modifications [24]. After the homogenization of the sample by mixing, 2 g was weighed in a 100 mL volumetric flask, followed by adding 40 mL of water and 4 mL of 2 M NaOH. The suspension was then vigorously shaken and 50 mL of 1 M phosphate buffer (pH 5.5) was added in order to lower the pH of the final solution to about pH 7. The suspension was made up to the mark with water and sonicated for 10 min in an ultrasonic bath. Dilutions of 20-fold with water were used for the quantification of the vitamins. The solution was filtered through a 0.22 µm Millipore syringe before analysis. Analyses were performed in duplicate.

#### 2.2.5. Standard Preparation

The multi-vitamins stock solution was prepared by weighing in a 100 mL volumetric flask 5 mg of vitamin B12; 12.5 mg of vitamin B2; 25 mg each of vitamins B1, B6 and B3. Forty millilitres (40 mL) of water was then added and the solution was shaken vigorously before adding 4 mL of 2 M NaOH. After complete dissolution of the vitamins, 50 mL of 1 M phosphate buffer (pH 5.5) was added and the solution made up to the mark with water. Stock standard solutions were prepared daily. Different concentrations of the standards were injected into the HPLC to obtain the peak areas. Peak areas were plotted against concentration for each vitamin to make specific calibration curves. The Shimadzu LC-2010CHT HPLC system was used with conditions according to Moreno et al. [25]. A volume of 20 µL for each sample was injected into the HPLC equipped with a C18 reversed-phase column (250 × 4.6 mm, 4 μm), 0.05 M ammonium acetate (solvent A)–methanol (solvent B) 92.5:7.2 as mobile phase at 1 mL/minute flow rate. A diode array detector was used to scan from 200 to 500 nm and LC-solutions software (Shimadzu, Japan) was used to integrate the peak areas for each vitamin. After the run, the peak area of each unknown sample was obtained, and concentrations were calculated using the calibration curves.

#### 2.2.6. Quality Control

To monitor the performance and reproducibility of the analytical procedures for the analysis of the B-vitamins, we included quality control samples spiked with known amounts of standards for each batch of samples on a daily basis. The means, standard deviations (SD) and coefficient of variation (CV%) for duplicate samples were calculated to ensure that the values were within the acceptable limits ≤10%. 

### 2.3. Characterization of Microbial Composition of Traditonal Fermented Foods Mabisi and Munkoyo

#### 2.3.1. Total Genomic DNA Extraction

Sample DNA was extracted from *Mabisi* and *Munkoyo* samples using the method by Schoustra et al. [13] Briefly, for *Munkoyo*, after eliminating large particles from 1 mL of product samples, they were spun down at high speed and the pellet was retained after discarding the supernatant. Then 500 µL TESL (25 mM Tris, 10 mM EDTA, 20% sucrose, 20 mg/mL lysozyme) and 10 µL mutanolysin solution (in water at 1 U/µL) were added, followed by incubation at 37 °C for 60 min with slight shaking. GES reagent (5 M guanidium thiocyanate, 100 mM EDTA, 0.5% sarkosyl) amounting to 500 µL was added, cooled on ice for 5 min and 250 µL of cold ammonium acetate solution (7.5 M) was added followed by gentle mixing. The mixture was held on ice for 10 min, spun down and the supernatant was removed. The samples were purified by mixing with chloroform-2-pentanol mix (chloroform and 2-pentanol 24:1 ratio) by adding 1:1 to the supernatant and the mixture was centrifuged to obtain the supernatant. Phenol-chloroform purification was performed by adding equal volume of phenol (i.e., tris-saturated phenol-chloroform-isoamyl ethanol in a ratio of 24:25:1) to the supernatant, vortexed for a few seconds, spun for 2 min at 12,000 rpm 4 °C and the supernatant was transferred to a fresh tube. An equal volume of chloroform was added to the supernatant, vortexed for a few seconds, spun 2 min at 12,000 rpm and 4 °C and the supernatant was transferred to a fresh tube. An amount of 2.5 volumes 100% ethanol was added, vortexed and precipitated the DNA at −80 °C for 3 h. Subsequently, samples were spun for 20 min at 12,000 rpm and 4 °C; the supernatant was removed by aspiration. DNA was washed by adding 1 mL cold 70% ethanol, spun for 10 min at 12,000 rpm and 4 °C; the supernatant was removed by aspiration and the DNA pellet was air-dried for 10 min at room temperature. The DNA was dissolved in 10 mM Tris treated with RNAse (10 mM Tris, bring to pH 8.0 with HCl; 1 mM EDTA; RNAse 20 µL/mL) and stored at −20 °C.

For the milk-based product (*Mabisi*), the DNA extraction protocol was performed as follows: into a 1.5 mL microcentrifuge tube was added 1 mL of *Mabisi* that was centrifuged at 13,000× *g* for 2 min to pellet the cells and remove the supernatant. The cells were re-suspended in a solution containing 64 µL of a 0.5 M EDTA solution, 160 µL of Nuclei Lysis Solution (Promega), 5 µL RNAse (10 mg/mL), 120 µL lysozyme (10 mg/mL) and 40 µL proteinase E (20 mg/mL) and incubated for 60 min at 37 °C. Ammonium acetate (5 M) 400 µL was added and cooled on ice for 15 min before being spun down at 13,000× *g* for 10 min. The supernatant containing the DNA was transferred to a fresh 1.5 mL microcentrifuge tube and a phenol-chloroform DNA purification was performed as described for *Munkoyo*.

#### 2.3.2. 16SrRNA Amplicon Sequencing of DNA Samples and Analysis of Sequence Data

The Company LGC Genomics GmbH (Berlin, Germany) conducted 16S rRNA gene analysis of bacterial communities in metagenomic DNA samples using the illumina MiSeq V3. Using an analysis pipeline [26] based on qiime software [27], the 25 samples collected from the producers in Mkushi were analyzed. Firstly, the forward and reverse reads were joined in one fastq sequence (join_paired_ends.py, minimum overlap 10 nucleotides). Then primers were removed from both ends and reads were quality trimmed using cutadapt (minimum length 400, minimum quality 20, [28]). With uchime, chimeric reads were removed (using blast against the “gold” database, [29]). Then the sequences were given identifier names by a custom awk script (similar to split_libraries.py). The command pick_open_reference_otu.py, at 0.95 similarity was used to cluster Operational Taxonomic Units (OTUs), to produce an OTU table and to assign taxonomy. From the OTU table produced, the minimum number of sequences per sample was determined and OTU tables were made by using multiple_rarefactions.py, to 15,000 sequences. Then alpha diversity and beta diversity were determined, which produced the distance matrices that were used for jackknife clustering (upgma_cluster.py), from which a consensus tree was produced (consensus_tree.py). Bacterial diversity by total effective sequence reads, OTU numbers, Chao1, and the Faith’s phylogenetic diversity (PD_Whole_Tree) were used to evaluate and compare diversity and richness of the communities among different samples, i.e., between and within samples taken per product type.

### 2.4. Statistical Analysis

R statistical package version 3.5.0 (version 3.3.1, R Foundation for Statistical Computing, Vienna, Austria) and IBM SPSS statistics version 25 (SPSS Inc., Chicago, IL, USA) were used to analyze the data. The nutrition composition data was presented as means with standard deviation (SD) and percentage coefficient of variation (%CV). A Student *t*-test and ANOVA as group tests were performed and principal components analysis (PCA) was carried out to determine variation in the samples. The nutritional variables were each categorized into three classes (low, medium, high) based on the percentage they could contribute to the estimated average requirement (EAR). The values that were able to contribute less than 20% of EAR were taken as low, for a contribution between 20% and 50% they were taken as medium, and for contribution of 50% and above then they were regarded as high.

For this study, we focused on the bacterial composition alone, since earlier work has shown that yeasts and other eukaryotes are usually present at an abundance below 1% [13]. The alpha diversity indices Faith’s phylogenetic diversity (PD) and Chao1 diversity richness were calculated for *Mabisi* and *Munkoyo* samples. Comparisons of diversity between *Mabisi* and *Munkoyo* samples were done using t-test of Chao1 diversity index. Then a non-parametric test, analysis of similarities (ANOSIM), was used to determine the impact of various independent variables including product type, fermentation vessel, and categories of nutritional parameters on the dependent variable microbial community composition. This analysis shows whether or not classifying the samples in distinct groups explains significant parts of the variation in microbial community composition between the samples—in our cases, the distinct groups are defined based on product and processing variables as well as on nutritional variables. 

## 3. Results

For our survey, 12 *Mabisi* and 13 *Munkoyo* samples were collected in Mkushi from different processors, each processor producing only one of the two types of traditional fermented food products. The *Mabisi* producers were two males and ten females all using the Tonga processing method [14], whereas all the *Munkoyo* producers were females and all used the Kitwe processing method [17]. The fermentation vessels used by the local processors for *Mabisi* were small plastic bottles (83%) and plastic buckets (17%), whereas for *Munkoyo* it was mostly calabashes (62%) and metal drums (38%). Samples were analyzed for their nutritional and microbial composition.

### 3.1. Nutritional Analyses

The results of the proximate analysis are in Table 1, and the levels of vitamins and minerals are in Table 2. Statistical tests comparing the results found for *Mabisi* and *Munkoyo* for each parameter are in Appendix A. The quality control revealed that all duplicate samples were below the acceptable range of 10% CV for AAS and HPLC. *Mabisi* samples on average had a moisture content between 85% and 90%, with two exceptions with a moisture content of 70%, while *Munkoyo* samples had a moisture content mostly between 90% and 95%. In comparison to the other *Mabisi* samples, the two *Mabisi* samples with lower moisture content showed high values of other proximate composition parameters and lower vitamin B2, vitamin B3 and calcium content, resulting in overall higher standard deviations around the mean over all samples for *Mabisi* than for *Munkoyo*. The pH for *Mabisi* on average is one pH unit higher than that for *Munkoyo*.

The % of the contribution to the EAR for women aged 19–50 years old for selected nutrients are shown in Table 3. One serving (183 g) of *Mabisi* would contribute mostly to the EAR of vitamin B2 (27%), calcium (22%), protein (18%), zinc (15%) but less than 10% of the EAR for the other B-vitamins and iron. One serving of *Munkoyo* would contribute to less than 10% of EAR for each of the nutrients. For each individual nutritional parameter that we measured, values are higher for *Mabisi* than for *Munkoyo* except for vitamin B1, vitamin B3 and vitamin B6.

Figure 1 shows a principle component analysis (PCA) to highlight what factors contribute to the variation in nutritional parameters between the samples of the two product types. Principal components 1 and 2 together explained about 77.4% of the variation between the samples. Principal component 1 (PC1) explains 65.6% of all variation with nutritional parameters high in positive loadings including calcium, ash, vitamin B2, vitamin B12, crude protein, carbohydrates, iron, energy, pH, crude fat, vitamin B3 and moisture (with a negative loading; see Appendix A). Principle component 2 (PC2) explains 11.8% of variation with variables high in loadings including vitamin B1 and vitamin B6. The PC-analysis clearly separates *Mabisi* and *Munkoyo* samples showing differences between the two products (Figure 1**)**. The *Munkoyo* samples are spread along PC2 and the variation is explained by vitamins B1 and B6 with higher loadings as in Table 1. Imposed on the graph are the nutritional parameters with vitamins B1, B3 and B6 (indicated by the red lines) separated away from the rest as they were not different between the two traditional fermented foods *Mabisi* and *Munkoyo*; moreover, moisture clustered with *Munkoyo* samples as it was higher in *Munkoyo* than *Mabisi*; other nutritional parameters are clustered with *Mabisi* samples as they were higher in *Mabisi* samples than in *Munkoyo* samples. The nutritional parameters are all aligned around zero for PC2 except for vitamins B1, B3 and B6, which were the reason why *Munkoyo* samples were differentiated along PC2. The vitamins B1, B3 and B6 were similar between *Mabisi* and *Munkoyo* samples but not the other nutritional parameters. This difference was confirmed using a *t*-test with results indicating that there was no statistically significant difference between *Mabisi* and *Munkoyo* for these vitamins (Appendix A). 

### 3.2. Microbial Analyses

Microbial community composition for each sample of *Mabisi* and *Munkoyo* as determined by non-culture-based methods are shown in Figure 2. In total, 1826 distinct bacterial types (Operational Taxonomic Units or OTUs) were found in all samples of which most were identified as either *Lactobacillus*, *Lactococcus*, *Streptococcus, Enterobacter*, *Klebsiella* or *Acetobacter.* Since even within one species taxonomic variation exists, different OTUs can be identified as the same species, yet each OTU does represent a unique bacterial type [32]. 

Two diversity indices were calculated to describe the microbial communities in the samples. The alpha diversity indices, Faith’s phylogenetic diversity (PD) and Chao1 were calculated for *Mabisi* and *Munkoyo* samples (Figure 3 and Appendix A). The Chao1 was different for samples of the different traditional fermented foods *Mabisi* and *Munkoyo* (*t*-test, t(23) = −7.18, *P* < 0.001). Based on Faith’s PD, there was no difference in microbial diversity between *Mabisi* and *Munkoyo* samples.

In the analysis of similarity (Table 4) assessing which variables contribute to the observed variation in microbial composition between the samples, we included two processing variables: the type of products (two categories, *Mabisi* and *Munkoyo*) and the type of fermentation vessel used (four categories). A *P*-value <0.05 indicates that the variable explains significant amounts of variation in the microbial community composition. Both processing variables explained significant parts of the variation in microbial profiles. We further included the categorical data of seven nutritional parameters for which sufficient variation exist to allow the statistical test; these were protein, fat, water soluble vitamins and minerals. Except for vitamin B1 and vitamin B3, the categorization of nutritional variables explained significant parts of the variation in the microbial community composition. 

## 4. Discussion

The aim of this study was to characterize the nutritional composition and microbial community composition of two traditional fermented foods, *Mabisi* that is based on raw milk and *Munkoyo* that is based on maize. The results in this study clearly showed that the two products were different with respect to the nutritional parameters and the microbial community composition.

### 4.1. Nutritional Composition

*Mabisi* was found to have higher nutritional values for crude protein, fat and carbohydrates than *Munkoyo*. The difference in nutritional composition between *Mabisi* and *Munkoyo* can be mainly attributed to the use of milk as the raw material for *Mabisi* and maize as the raw material for *Munkoyo*. Milk is a rich animal source of protein and fat, while the main component in maize is starch [33,34]. Among samples of the same product type, variation in nutritional composition is higher for *Mabisi* samples than for *Munkoyo* samples. This is mainly caused by two *Mabisi* samples that have a lower moisture content than the other samples. This lower moisture content may be caused by the removal of whey during the fermentation process. While specific processors in our study did not mention whey removal, other studies have found that several processors remove whey, reducing the volume of the processing batch by 30% [14].

In Zambia, fresh milk (unfermented) is rarely consumed due to high prevalence of lactose intolerance, which is estimated at 70–90% in the Zambian population [35]. During fermentation, most of the lactose, which at the beginning is an antinutritional factor, is converted into lactic acid and other compounds [4]. In the Zambian context, this makes *Mabisi* more nutritious than fresh milk. It was expected that *Mabisi* would have had a higher concentration of B-vitamins considering that it is made from milk and *Munkoyo* is made from maize which has low levels of most B-vitamins. We found however that both products are a source of B-vitamins, which could be attributed to the fermenting bacteria which have previously been shown to produce B-vitamins [3]. *Mabisi* was higher in calcium, iron and zinc and regular consumption in combination with other local foods could help to increase intake of these micronutrients. This was also reflected when one serving of *Mabisi* for an adult woman was considered to contribute higher amounts of calcium and zinc and also vitamin B2 and protein to the estimated average requirements. It can be said therefore that *Mabisi* would be a good source of nutrients for inclusion in the food-based dietary guidelines [36]. A recent study using 24 h recalls to measure micronutrient intakes of lactating women in rural Zambia found inadequate intakes, especially of vitamin B3, vitamin B12 and iron [37]. These B-vitamins are of interest to our study since microbial activity could increase their levels in the final products. Furthermore, since raw milk is not consumed that much due to lactose intolerance, the promotion of *Mabisi* could have a positive impact on iron intakes. This positive impact may apply more broadly to other dairy based traditional fermented foods in the region [18,38]. 

### 4.2. Microbial Community Composition

Our results showed that the microbial communities in the product samples consist of three to eight distinct bacterial types (Figure 2). Several different bacterial types belong to the same species [32]. Previous studies have shown that bacterial densities in the final products are typically around 10^8^ cfu/g [13]. The microbial community composition in *Mabisi* samples was most abundant in *Streptococcus*, *Enterobacter* and *Lactococcus* species, while the microbial community composition in *Munkoyo* samples was most abundant in *Lactococcus* and *Lactobacillus* species, which is consistent with other studies [13,17]. The microbial communities in the products are dominated by lactic acid bacteria, whose growth resulted in a low pH, enhancing food safety properties and shelf-life. For *Mabisi*, the final pH was around 4.1 and for *Munkoyo* the final pH was around 3.2, which is in line with previous studies [13]. Products at a pH below 4.5 are generally considered to be protected against microbial pathogen proliferation [39]. The pH values we found for *Munkoyo* were consistently well below this safety threshold. However, for one *Mabisi* sample, we found a pH of 4.6, highlighting that during *Mabisi* processing, the pH level could be a safety concern. 

The lactic acid bacteria are also regarded as healthy bacteria that may enable shifts in gut microbiota composition towards a more healthy composition. *Mabisi* had a slightly higher diversity as shown by the diversity indices that we calculated. This could be attributed to the fact that raw milk contains a wider diversity in substrates supporting a wider range of bacterial types, especially in that raw milk contains more protein. More complex substrates are known to support more diverse species communities [40]. The Chao1 diversity index showed higher diversity in *Mabisi* than *Munkoyo*, whereas Faith’s phylogenetic diversity index was the same between the two products. This may be caused by the fact that the Faith’s phylogenetic diversity index uses branch lengths for assigning diversity metrics, which cannot separate lactic acid bacteria with the same level of discriminatory detail [41]. The Chao1 index on the other hand is an estimator based on the abundance of species taking into account the rare species [42]. Alpha diversity index Chao1 found in *Mabisi* samples (ranged from 206 to 471) was higher than what Shangpling et al. (2018) found (Chao1 ranged from 90 to 138) for the Indian naturally fermented milk product [43]. However, our results were comparable with what Liu Xiao-Feng et al. (2015) found for a Chinese traditional fermented goat milk (Chao1 ranged from 166 to 640) [44].

### 4.3. Factors that Affect Microbial Community Composition

It is thought that the composition of species’ communities depends on external selection pressures that lead to a process of species sorting [45,46]. In our study, the main contrast in external selection were the raw materials and fermentation vessels used for fermentation. As expected, our results show a marked difference in microbial composition between *Mabisi* (based on milk) and *Munkoyo* (based on maize), which however is in slight contrast with earlier work [13]. This earlier work had compared various microbial communities from *Mabisi* and *Munkoyo* samples collected at various distant geographic locations and did not control for a processing method. They found that microbial communities collected at the same location had similar microbial communities, regardless of the product type (*Mabisi* or *Munkoyo*), suggesting that it is geographical location rather than raw materials that most significantly affects microbial community structure [13]. The present study was performed in a more systematic way, focussing on one processing method per product type and one geographical location. In our study, the differences between the microbial communities of the two products could be due to variations in other determinants known to affect microbial community composition, in particular the fermentation vessel used, which indeed came out as a significant factor explaining variation in microbial communities, and the level of back-slopping. It has been established that, for example, back-slopping, which is the transfer of a small fraction of the previous product into fresh raw material [47], ensuring the transfer of microbial communities underlying the fermentation helps shape microbial communities from batch to batch. In Zambia, back-slopping is usually done using a calabash as a fermenting vessel which is not washed to preserve some starter cultures that is used for the next fermentation and in our study 68% of the *Munkoyo* producers had used the calabash and none for *Mabisi*. This could imply that most of the *Mabisi* producers in this study area rely on the spontaneous fermentation method. This could mean that indeed environment played a role in shaping the microbial composition but also a fermenting substrate could play its part because of the similarities in samples of the same type which is in agreement with what has been found before [8,9].

We found a correlation between levels of various nutrients (levels of protein, fat, B-vitamins and calcium) and a variation in microbial community structure (Table 4). Our experimental design does not allow to distinguish whether different levels of nutrients in the raw materials affected the microbial community composition, or the other way around—that composition of microbial communities affects metabolic activity, resulting in some final products to have higher levels of nutrients. We hypothesize that levels and types of protein, carbohydrates and maybe calcium in raw materials are a selective force in driving species composition since these are nutrients that are directly used by a wide range of micro-organisms. Moreover, different micro-organisms have different requirements and capabilities to metabolise these substrates. On the contrary, the microbial community composition may affect the final levels of B-vitamins, since several B-vitamins are known to be produced by bacteria and are added to the raw materials by fermentation [48,49]. For instance, *Lactococcus lactis* has been found to produce significant amounts of riboflavin during fermentation [50]. Our finding suggests that research to determine the levels of enrichment with B-vitamins by the micro-organisms present in *Mabisi* and in *Munkoyo* fermentation is worthwhile. This could be done in controlled laboratory experiments using defined mixtures of bacteria isolated from *Mabisi* and *Munkoyo* and measuring levels of B-vitamins before and after proliferation of the bacteria. This future work could also include experiments with defined bacterial communities to identify the specific micro-organisms that are responsible for B-vitamin production. This could further be extended to other studies on the functionality of microbial communities, for instance in the removal of mycotoxins from maize during fermentation [51]. In the present study, we collected product samples from producers and did not perform the fermentation ourselves. The present work could not permit us to carry out baseline nutrition analysis on the raw materials before fermentation so that we could attribute any changes in microbial composition to the difference between baseline and after fermentation. 

## 5. Conclusions and Significance

This study documented nutritional composition of traditional fermented foods *Mabisi* and *Munkoyo* with *Mabisi* having higher nutrient values than *Munkoyo* except for vitamins B1, B3 and B6. We determined the composition of micro-organisms that are present in *Mabisi* and *Munkoyo*. *Mabisi* has an advantage over *Munkoyo* for is consumers in that it has a greater impact on nutrient intake. The increase in B-vitamins and the possible probiotic effect of *Munkoyo* also makes it a product that is useful for regular consumption for an improvement in dietary diversity. We assessed and found that differences in microbial communities correlated to differences in nutritional content. Our study thus provides unique data on the nutritional composition of two traditional fermented foods that is essential for the planning of nutritional programmes in Zambia. It provides a general outlook on the importance of understanding how microbial activity adds to the nutritional value of fermented products. 

Our study is a formal demonstration that a locally produced fermented food, especially *Mabisi*, can contribute to achieving improved nutrient intake of various important macro- and micro-nutrients. For many of the locally available foods such as *Mabisi* and *Munkoyo*, nutritional data are lacking, impeding their consideration for inclusion in food-based dietary guidelines. Therefore, the data generated in this study will be useful for inclusion in the food-based guidelines. We recommend more research to include a determination of the nutritional composition of raw materials and end-products of fermentation to quantify the addition of nutrients by fermenting microbes and to conduct a more genomic analysis for B-vitamin production. Furthermore, other recent work has shown that a variation in *Mabisi* and *Munkoyo* processing methods have an impact on microbial community composition. Based on this current study, this variation in microbial community structure may also impact nutritional composition. Thus, the inclusion of other processing types of *Mabisi* and *Munkoyo* than the ones used in this study is also recommended. Finally, our work could be expanded by adding measures of bioavailability of the nutrients within the diets of consumers by determining molar ratios of phytate to zinc, iron and calcium.

## Figures and Tables

**Figure 1 nutrients-12-01628-f001:**
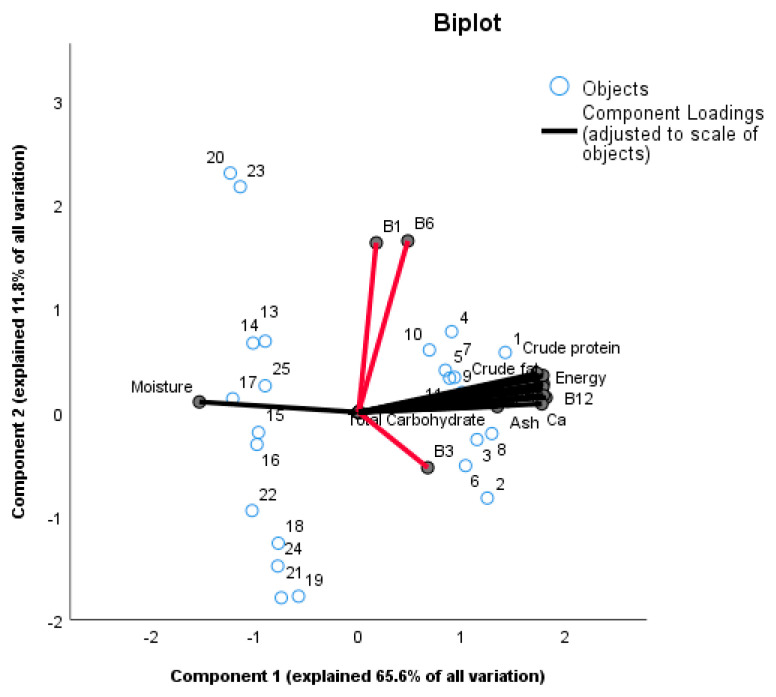
Principal Components Analysis (PCA) bi-plot with Optimal Scaling of *Mabisi* (numbers 1 to 12) and *Munkoyo* (numbers 13 to 25) samples are indicated as blue circles to determine variation in nutritional parameters within and in between sample types. PC1 explained 65.6% of all variation while PC2 explained 11.8% of all variation giving a total of 77.4% variation explained by the two components.

**Figure 2 nutrients-12-01628-f002:**
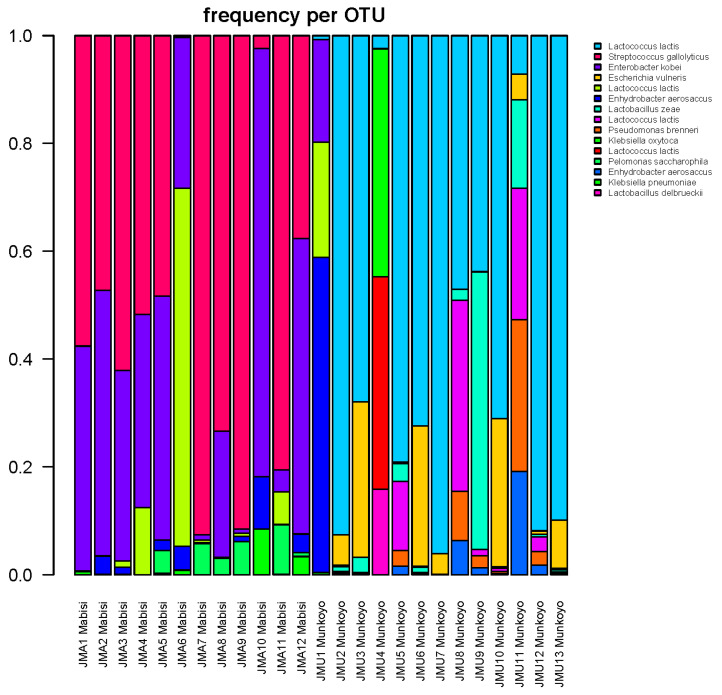
Relative abundance of bacterial types that constitute the bacterial communities in traditional fermented foods *Mabisi* and *Munkoyo*; samples with the y-axis represent relative abundance and the x-axis represents the different samples. Colors in the bars show different Operational Taxonomic Units (OTUs), the legend shows the most likely BLAST identity of that OTU. Multiple OTUs can be identified as the same species.

**Figure 3 nutrients-12-01628-f003:**
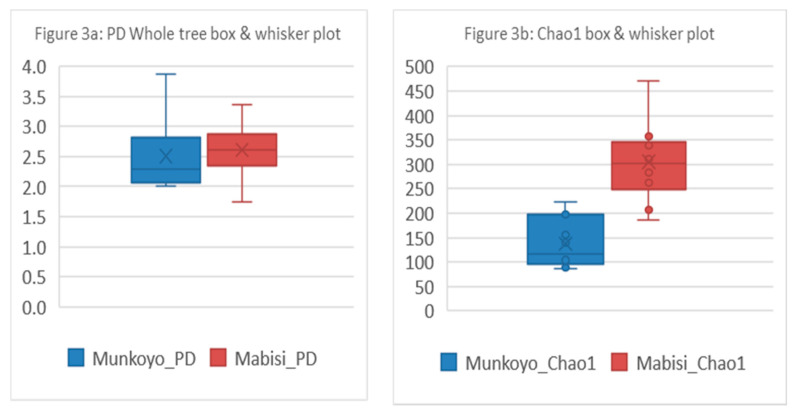
Comparison of alpha diversity indices of microbial communities in traditional fermented foods *Mabisi* and *Munkoyo* samples based on Faith’s phylogenetic diversity (PD) and Chao1. The blue box plots indicate *Munkoyo* and red box plots indicate *Mabisi*. (**a**) shows Faith’s Phylogenetic diversity index of alpha diversity and (**b**) shows plots for Chao1 index of alpha diversity.

**Table 1 nutrients-12-01628-t001:** Proximate analysis and pH of traditional fermented foods *Mabisi* and *Munkoyo.*

Sample	Moisture			Dry Matter			Ash			Fibre			Crude Protein			Crude Fat			Total Carbohydrate			Energy			pH
	%	SD	CV%	%	SD	CV%	%	SD	CV%	%	SD	CV%	%	SD	CV%	%	SD	CV%	%	SD	CV%	Kcal/100 g	SD	CV%	
*Mabisi*																									
1	67.4	±0.378	0.6	32.6	0.400	1.3	0.47	±0.023	4.9	0	±0.000	0	4.68	±0.045	1.0	4.2	±0.181	4.3	23.28	±0.567	2.4	149.67	±0.747	0.5	4.1
2	85.2	±5.622	6.6	14.8	0.033	0.3	0.65	±0.006	0.9	0	±0.000	0	3.17	±0.006	0.2	3.37	±0.275	8.2	7.6	±5.427	71.4	73.38	±23.537	32.1	4.13
3	88.1	±0.172	0.2	11.9	0.182	1.5	0.73	±0.01	1.4	0	±0.000	0	3.34	±0.175	5.2	3.83	±0.404	10.5	4.04	±0.546	13.5	64.03	±2.109	3.3	4.65
4	89.3	±0.323	0.4	10.8	0.086	0.8	0.71	±0.061	8.6	0	±0.000	0	3.45	±0.365	10.6	3.23	±0.308	9.5	3.35	±0.842	25.1	56.21	±0.994	1.8	4.1
5	89.6	±0.246	0.3	10.4	0.335	3.2	0.62	±0.015	2.4	0	±0.000	0	5.27	±0.170	3.2	3.08	±0.534	17.3	1.39	±0.38	27.3	54.34	±3.066	5.6	4.1
6	87.1	±0.530	0.6	12.9	0.531	4.1	0.72	±0.021	2.9	0	±0.000	0	4.46	±0.085	1.9	4.67	±0.117	2.5	3.08	±0.54	17.5	72.15	±1.466	2.0	4.05
7	88.6	±0.068	0.1	11.4	0.196	1.7	0.71	±0.02	2.8	0	±0.000	0	4.3	±0.091	2.1	4.07	±0.563	13.8	2.32	±0.566	24.4	63.08	±3.165	5.0	4.1
8	73.9	±0.559	0.8	26.1	0.580	2.2	0.63	±0.01	1.6	0	±0.000	0	3.38	±0.221	6.5	6.98	±0.472	6.8	15.08	±0.795	5.3	136.63	±1.041	0.8	4
9	89.5	±0.168	0.2	10.5	0.168	1.6	0.71	±0.01	1.4	0	±0.000	0	2.83	±0.179	6.3	4.5	±0.532	11.8	2.46	±0.551	22.4	61.68	±3.306	5.4	4.05
10	88.9	±0.246	0.3	11.1	0.374	3.4	0.68	±0.025	3.7	0	±0.000	0	4.55	±0.26	5.7	3.75	±0.462	12.3	2.11	±0.418	19.8	60.39	±1.725	2.9	4
11	89.1	±0.260	0.3	10.9	0.290	2.7	0.74	±0.049	6.6	0	±0.000	0	4.07	±0.071	1.7	3.19	±0.064	2.0	2.93	±0.262	8.9	56.73	±1.493	2.6	4.1
12	87.4	±0.355	0.4	12.6	0.259	2.1	0.65	±0.025	3.8	0	±0.000	0	2.93	±0.268	9.1	3.85	±0.137	3.6	5.18	±0.347	6.7	67.1	±1.379	2.1	4.1
Mean	85.34	3.61	4.2	14.67	1.81	12.3	0.67	0.074	11	0	0		3.87	0.786	20.3	4.06	1.053	25.9	6.07	6.57	108.2	76.28	31.901	41.8	4.12
*Munkoyo*																									
1	95.7	±0.000	0.0	4.3	0.07	1.7	0.19	±0.012	6.3	0.72	±0.062	8.6	0.39	±0.045	11.5	0.56	±0.231	41.3	2.46	±0.122	5.0	16.48	±1.432	8.7	3.1
2	92.5	±0.457	0.5	7.5	0.026	0.4	0.19	±0.006	3.2	0.69	±0.023	3.3	0.79	±0.082	10.4	1.02	±0.180	17.6	4.78	±0.396	8.3	31.49	±2.184	6.9	3.05
3	95.9	±0.000	0.0	4.1	0.001	0.1	0.16	±0.025	15.6	0.85	±0.121	14.2	0.51	±0.087	17.1	0.63	±0.164	26.0	1.99	±0.025	1.3	15.67	±1.206	7.7	3.1
4	95.4	±0.081	0.1	4.6	0.116	2.5	0.14	±0.006	4.3	0.7	±0.006	0.9	0.36	±0.017	4.7	0.36	±0.092	25.6	3.08	±0.056	1.8	17	±0.730	4.3	3.2
5	95.8	±0.044	0.0	4.2	0.05	1.2	0.11	±0.012	10.9	0.67	±0.031	4.6	0.23	±0.031	13.5	0.85	±0.015	1.8	2.3	±0.040	1.7	17.8	±0.110	0.6	2.9
6	92.5	±0.105	0.1	7.5	0.148	2	0.16	±0.031	19.4	0.5	±0.006	1.2	0.49	±0.045	9.2	0.93	±0.432	46.5	5.39	±0.408	7.6	31.85	±2.423	7.6	3.35
7	90.2	±0.062	0.1	9.8	0.086	0.9	0.16	±0.021	7.5	0.68	±0.006	0.9	0.21	±0.042	20.0	0.76	±0.071	9.3	7.96	±0.021	0.3	39.52	±0.542	1.4	3.4
8	94.5	±0.061	0.1	5.5	0.084	1.5	0.1	±0.006	6.0	0.41	±0.223	54.4	0.18	±0.080	44.4	0.86	±0.108	12.6	3.96	±0.267	6.7	24.32	±1.580	6.5	3.25
9	89	±0.050	0.1	11	0.034	0.3	0.1	±0.006	6.0	0.49	±0.221	45.1	0.26	±0.006	2.3	0.75	±0.092	12.3	9.45	±0.146	1.5	45.59	±1.353	3.0	3.3
10	95.5	±0.296	0.3	4.5	0.418	9.4	0.04	±0.000	0.0	0.57	±0.107	18.8	0.15	±0.042	28.0	0.95	±0.128	13.5	2.75	±0.029	1.1	20.15	±1.391	6.9	3
11	94.6	±0.030	0.0	5.4	0.042	0.8	0.06	±0.020	33.3	0.61	±0.059	9.7	0.46	±0.099	21.5	0.74	±0.096	13.0	3.57	±0.053	1.5	22.82	±0.673	4.3	3.3
12	92.4	±0.355	0.4	7.6	0.446	6	0.11	±0.010	9.1	0.54	±0.012	2.2	0.35	±0.006	1.7	1.46	±0.070	4.8	5.14	±0.425	8.3	35.09	±1.063	3.0	2.9
13	94.4	±1.267	1.3	4.6	0.57	12.6	0.13	±0.000	0.0	0.77	±0.010	1.3	0.27	±0.012	4.4	0.72	±0.056	7.8	3.75	±1.239	33.0	22.55	±5.323	23.6	3.2
Mean	93.72	2.24	2.4	6.20	2.28	36.8	0.13	0.046	38.3	0.63	0.124	20.0	0.36	0.175	48.6	0.81	0.262	31.2	4.35	2.23	49.4	26.18	9.651	35.6	3.16

Notes: SD is the standard deviation, CV% is percentage coefficient of variation.

**Table 2 nutrients-12-01628-t002:** Content of selected vitamins and minerals in the samples of traditional fermented foods *Mabisi* and *Munkoyo.*

Sample	Vitamin B1			Vitamin B2			Vitamin B3			Vitamin B6			Vitamin B12			Calcium			Iron			Zinc		
	mg/100 g	SD	CV%	mg/100 g	SD	CV%	mg/100 g	SD	CV%	mg/100 g	SD	CV%	µg/100 g	SD	CV%	mg/100 g	SD	CV%	mg/100 g	SD	CV%	mg/100 g	SD	CV%
*Mabisi*																								
1	0.062	±0.002	3.2	0.133	±0.002	1.5	0.131	±0.008	6.1	0.017	±0.001	5.9	0.327	±0.018	5.5	58	±1.989	1.9	0.625	±0.017	2.7	0.748	±0.112	15
2	0.034	±0.003	8.8	0.151	±0.002	1.3	0.455	±0.008	1.8	0.016	±0.000	0	0.429	±0.016	3.7	104.8	±8.534	8.1	0.273	±0.013	4.8	0.625	±0.049	7.8
3	0.03	±0.002	6.7	0.127	±0.002	1.6	0.222	±0.013	5.9	0.017	±0.000	0	0.372	±0.014	3.8	116.5	±6.028	5.2	0.264	±0.010	3.8	0.85	±0.102	12
4	0.072	±0.016	22.2	0.136	±0.002	1.5	0.375	±0.016	4.3	0.019	±0.001	5.3	0.475	±0.018	3.8	109.6	±2.693	2.5	0.183	±0.015	8.2	0.778	±0.047	6
5	0.044	±0.002	4.5	0.129	±0.001	0.8	0.415	±0.012	2.9	0.021	±0.003	14.3	0.411	±0.018	4.4	106.2	±5.821	5.5	0.133	±0.015	11.3	0.586	±0.039	6.7
6	0.031	±0.001	3.2	0.151	±0.002	1.3	0.566	±0.005	0.9	0.023	±0.002	8.7	0.528	±0.016	3	107	±5.534	5.2	0.029	±0.002	6.9	0.458	±0.041	9
7	0.034	±0.002	5.9	0.148	±0.002	1.4	0.261	±0.004	1.5	0.023	±0.001	4.3	0.326	±0.004	1.2	104.8	±8.794	8.4	0.105	±0.012	11.4	0.789	±0.033	4.2
8	0.024	±0.001	4.2	0.126	±0.003	2.4	0.169	±0.005	3	0.022	±0.001	4.5	0.317	±0.010	3.2	68.5	±9.668	14.1	0.15	±0.014	9.3	0.588	±0.020	3.4
9	0.034	±0.003	8.8	0.131	±0.002	1.5	0.366	±0.004	1.1	0.026	±0.001	3.8	0.339	±0.024	7.1	110.1	±16.757	15.2	0.166	±0.011	6.6	0.684	±0.032	4.7
10	0.036	±0.001	2.8	0.121	±0.002	1.7	0.24	±0.012	5	0.028	±0.001	3.6	0.342	±0.014	4.1	96.2	±5.754	6	0.113	±0.010	8.8	0.648	±0.050	7.7
11	0.023	±0.002	8.7	0.132	±0.005	3.8	0.257	±0.007	2.7	0.03	±0.001	3.3	0.342	±0.013	3.8	101.8	±8.284	8.1	0.124	±0.005	4	0.74	±0.032	4.3
12	0.054	±0.005	9.3	0.101	±0.075	74.3	0.494	±0.008	1.6	0.017	±0.001	5.9	0.52	±0.018	3.5	98.5	±2.535	2.6	0.096	±0.010	10.4	0.609	±0.026	4.3
Mean	0.04	0.015	38.2	0.132	0.014	10.6	0.329	0.136	41.4	0.022	0.005	21.1	0.394	0.077	19.6	98.5	17.4	17.7	0.188	0.154	81.7	0.675	0.110	16.4
*Munkoyo*																								
1	0.057	±0.003	5.3	0.045	±0.002	4.4	0.34	±0.039	11.5	0.019	±0.001	5.3	0	±0.000		2.4	±0.344	14.3	0.067	±0.004	6	0.267	±0.002	0.7
2	0.043	±0.003	7	0.031	±0.001	3.2	0.244	±0.008	3.3	0.022	±0.001	4.5	0	±0.000		3.9	±0.105	2.7	0.092	±0.004	4.3	0.328	±0.003	0.9
3	0.026	±0.001	3.8	0.045	±0.001	2.2	0.097	±0.007	7.2	0.014	±0.001	7.1	0	±0.000		2.9	±0.081	2.8	0.047	±0.005	10.6	0.318	±0.002	0.6
4	0.032	±0.002	6.3	0.056	±0.002	3.6	0.082	±0.004	4.9	0.016	±0.001	6.3	0	±0.000		5.1	±0.098	1.9	0.023	±0.002	8.7	0.167	±0.003	1.8
5	0.034	±0.002	5.9	0.027	±0.001	3.7	0.24	±0.006	2.5	0.018	±0.001	5.6	0	±0.000		3	±0.055	1.8	0.056	±0.002	3.6	0.243	±0.003	1.2
6	0.022	±0.001	4.5	0.037	±0.001	2.7	0.301	±0.022	7.3	0.01	±0.001	10	0	±0.000		2.7	±0.154	5.7	0.019	±0.002	10.5	0.257	±0.003	1.2
7	0.022	±0.001	4.5	0.039	±0.001	2.6	0.498	±0.020	4	0.006	±0.001	16.7	0	±0.000		8.7	±0.706	8.1	0.013	±0.001	7.7	0.269	±0.002	0.7
8	0.074	±0.002	2.7	0.028	±0.002	7.1	0.094	±0.011	11.7	0.034	±0.002	5.9	0	±0.000		0.9	±0.067	7.4	0.026	±0.001	3.8	0.401	±0.002	0.5
9	0.013	±0.001	7.7	0.025	±0.001	4	0.36	±0.019	5.3	0.004	±0.001	25	0	±0.000		2.9	±0.123	4.2	0.05	±0.002	4	0.252	±0.003	1.2
10	0.023	±0.001	4.3	0.036	±0.001	2.8	0.059	±0.005	8.5	0.008	±0.001	12.5	0	±0.000		3.3	±0.214	6.5	0.09	±0.002	2.2	0.262	±0.003	1.1
11	0.068	±0.002	2.9	0.053	±0.002	3.8	0.197	±0.016	8.1	0.035	±0.002	5.7	0	±0.000		2.7	±0.087	3.2	0.033	±0.001	3	0.182	±0.001	0.5
12	0.017	±0.002	11.8	0.032	±0.002	6.3	0.227	±0.019	8.4	0.005	±0.001	20	0	±0.000		6.9	±0.331	4.8	0.129	±0.003	2.3	0.326	±0.003	0.9
13	0.037	±0.002	5.4	0.044	±0.002	4.5	0.235	±0.004	1.7	0.017	±0.002	11.8	0	±0.000		2.1	±0.133	6.3	0.058	±0.002	3.4	0.291	±0.002	0.7
Mean	0.036	0.019	53.9	0.038	0.01	25.9	0.229	0.127	55.6	0.016	0.01	62.6	0	0		3.654	2.106	57.6	0.054	0.034	62.6	0.274	0.062	22.6

Notes: SD is the standard deviation, CV% is percentage coefficient of variation.

**Table 3 nutrients-12-01628-t003:** The contribution of the traditional fermented foods *Mabisi* or *Munkoyo* to the Estimated Average Requirements (EAR) of selected nutrients for women 19–50 years old [30].

EAR ^a^	Women 19–50 Years Old	%EAR *Mabisi*	%EAR *Munkoyo*
Protein ^b^ (g)	40	18%	2.0%
Vitamin B1 (mg)	0.90	8.0%	7.3%
Vitamin B2 (mg)	0.90	27%	8.0%
Vitamin B3 (mg)	11	5.5%	4.0%
Vitamin B6 (mg)	1.1	4.0%	3.0%
Vitamin B12 (µg)	2.0	3.0%	0%
Calcium (mg)	833	22%	0.80%
Iron ^c^ (mg)	59	0.60%	0.20%
Zinc ^d^ (mg)	8.2	15%	6.0%

^a^ Units apply to the EAR for women 19–50 years old. ^b^ portion size was 183 g [19]. For protein no WHO EARs exist and 80% of the Population Reference Intake from the European Food and Safety Authority (EFSA) was used (80% × 0.83 g protein/kg body weight per day) [31]. ^c^ For iron, 5% bioavailability was assumed. ^d^ for zinc, low bioavailability was assumed.

**Table 4 nutrients-12-01628-t004:** Results of the analysis of similarity (ANOSIM) for impact on microbial composition of product type and fermentation vessel and various nutritional parameters for which sufficient variation exists among samples.

Variable	Anosim R	Number of Treatment Groups	*p*-Value
Product and processing variables			
Product type Mabisi or Munkoyo	0.816	2	<0.001
Fermentation vessel used	0.605	4	<0.001
Nutritional variables			
Protein	0.816	2	<0.001
Fat	0.78	3	<0.001
Vitamin B1	−0.0683	2	0.718
Vitamin B2	0.532	3	<0.001
Vitamin B3	−0.0374	2	0.436
Vitamin B12	0.816	2	<0.001
Calcium	0.772	3	<0.001

Notes: Variable, test statistic (R), number of treatment groups (# of Groups) and exact *p* value (P) are given, unless the *p* value was smaller than 0.001, which is indicated by <0.001.

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
