# Peer review of "Nutritional Composition and Microbial Communities of Two Non-alcoholic Traditional Fermented Beverages from Zambia: A Study of Mabisi and Munkoyo"

_nutrients, 2020, doi:10.3390/nu12061628_

Round 1

Reviewer 1 Report

Is an interesting study, but I would recommend that  some  aspects could be improved upon to strengthen the manuscript.

Author Response

Comments and Suggestions for Authors

Is an interesting study, but I would recommend that  some  aspects could be improved upon to strengthen the manuscript.

*** We thank the Reviewer for the time and effort spent on reviewing the manuscript and for the constructive feedback. We have incorporated the suggestions.

See comments in the pdf version of the paper.

Abstract: Change wording of Mabisi and Munkoyo to Fermented foods Mabisi and Munkoyo

*** We have added “traditional fermented foods” multiple times in the manuscript, ensuring that at least at the start of every section the reader is reminded that Mabisi and Munkoyo are traditional fermented foods. We have also added this information to all titles of tables and figures in the main manuscript. We opted not to provide this addition each time since then we would significantly increase our word count. We trust however that when reading the manuscript after our change, it is very clear that Mabisi and Munkoyo are fermented foods.

Line 123 Proximate analysis: Proximate analysis: Different proximate parameters in Munkoyo and Mabisi were determined using the methods of the Association of Official Analytical Chemists (AOAC). Here, t should indicate how this quantity has been calculated (183g).

*** The amount of 183g was calculated as an average of portion values used in another study on similar products. We have now cited the reference for this (Hotz, C., Palaniappan, U., Chileshe, J., Kafwembe, E., & Siamusantu, W. (2011). Nutrition survey in central and eastern provinces, Zambia 2009: Focus on vitamin A and maize intakes, and vitamin A status among women and children. Washington (DC): National Food and Nutrition Commission of Zambia, Lusaka, Zambia).

Line 267: Title of Table 1. delete, since Table 1 and the title are shown below.

*** The title has been deleted.

All tables and figures: In all Tables change Mabisi and Munkoyo to Traditional fermented food Mabisi and Munkoyo.

*** This change has been made to all titles of Figures and Tables in the manuscript.  

Line 271. Put notes parameters in the same order of appearance as in the table and at the end p values <0.05.

*** This has been corrected. We have added a separate Table with the results of the statistical analysis (Supplementary Table 1).

Throughout. Change numbering of the tables to correct order.

*** This have been corrected.

Table 4. Correct typos.

*** These have been corrected.

Line 322, in table. In Table 5 Need to indicate the units for the different nutritional parameters.

*** These have been added (the Table is now Table 3).

Line 352. Why are  iron and zinc results not included in the Table 4?

*** These variables have not been included in the statistical analysis since insufficient variation existed to allow for the test. This has been clarified in the text (Line 351-354).  

Table 6. Write Vitamin Bxx in stead of just Bx

*** This has been added to what is now Table 4, also in what is now Supplementary Table 3.

References. Adjust formatting to journal style: use short names of journals.

*** This have been corrected after installing a newer version of the EndNote output style for this journal.

Reviewer 2 Report

General comments

The present work concerns with the characterization of two traditional Zambian fermented food made with maize and milk. The topic of the work is interesting, many analyses were carried out to characterize these products and it is the first step for understanding the nutritional potentiality of these products. However, I think that the experimental part and the methodology were not properly conducted and described, and the work needs substantial major revisions. English must be considerably improved in all the section, please avoid the continuous use of “we” when describing the work carried out. Check the number of tables, since 4 (!) of them are wrong numbered and placed in the wrong order, and if you show a figure or table, you’re supposed to comment it. In addition, I don’t see the point of the comparison between the products, they are made with two different raw materials and I think that they should be commented separately. I found the conclusions related to microbial activity not consistent with the obtained data.

Introduction:

Some examples for English improvement

Line 42 “Not only are these foods” change in “These are food not only locally..”

Line 48 “on their diverse microbial community.”

Line 54 you were talking of general properties and then talk specifically of milk, keep staying general and more specific in the successive part.

Line 75 change between in among

Line 83 change between in among

M &M

Check the spaces between numbers and unit measure and report City and State of the company.

Line 100-102, why you asked their age and education degree? You should have asked  about the processes (time, temperature, backslopping ..)

Line 177-180 these part goes in the results or discussion

Line 246-249 Do not explain in M&M

pH was measured but it is not reported in M&M

Results

Line 253 -257 I don’t think these part about ages and education is useful for the results

Table 1 column Carbohydrates, percentage coefficient of variation is missing

Line 280 Table 3 is not the table you are referring to and should not be placed in this part of the results

Figure 1 in the caption you don’t have to describe the results (Line 293-302), Line 297: orange and green ellipses not reported

Table 4 you have to introduce table and this is commented later on the manuscript

Line 311 is not Table 5

Table 5 you should improve this table and comment it, I think you can add the EAR% values of Mabisi and Munkoyo to better understand their nutritional values instead of putting these data on table 1

Figure 2 why Lactococcus lactic is represented by four colors? I would suggest more descriptions of this figure. Do you have an idea of the species concentration as CFU/g? I think this is of interest since you explain some difference thanks to microbial activity.

Line 338 it is not Table 4 but 3, please adjust the order and place tables in the correct order

Line 347 I think you mean Table 4, why you chose those nutritional parameters?  And how for example protein and fat contribute to microbial community?

Table 6 This table has to be moved from this part

Discussion

I think that you should provide some information about the composition of maize and milk, in order to explain the Mabisi and Munkoyo differences and consider them as two separated products that you primarily characterize rather than compare. Moreover, we do not know anything about the fermented food process (except for the vessels) that play a role in their compositions.

Line 379  and 423 you should restate this sentence, some strains are able to produce these vitamins, and which species among those that you found could be able ?

Author Response

General comments

The present work concerns with the characterization of two traditional Zambian fermented food made with maize and milk. The topic of the work is interesting, many analyses were carried out to characterize these products and it is the first step for understanding the nutritional potentiality of these products. However, I think that the experimental part and the methodology were not properly conducted and described, and the work needs substantial major revisions. English must be considerably improved in all the section, please avoid the continuous use of “we” when describing the work carried out. Check the number of tables, since 4 (!) of them are wrong numbered and placed in the wrong order, and if you show a figure or table, you’re supposed to comment it.

*** We thank the Reviewer for the constructive comments that have allowed us to substantially improve the manuscript. We agree that in the previous version of the manuscript, the reference in the text to the Figures and Tables was too brief. These references to Tables and Figures have now expanded, mainly in the results section, for instance see (Line 282-285 and Line 302-321). The use of “we” has been limited throughout the manuscript. Somehow the numbering of the Tables was wrong, this has been corrected.

In addition, I don’t see the point of the comparison between the products, they are made with two different raw materials and I think that they should be commented separately. I found the conclusions related to microbial activity not consistent with the obtained data.

*** We chose to compare the two products mainly for their microbial community composition since previous work had suggested that the microbial communities in both products have a highly similar composition – which was a striking and unexpected result. This point is now highlighted in the introduction and covered more explicitly in the discussion section. We agree that for the nutritional composition, it was to be expected that differences exist between the two product types since they have such different raw materials. The comparisons of nutritional differences between the product types has been removed from the Tables where we present these results. Since some readers may still find this interesting and also since not for all nutrients analysed a difference was found, we do still report on the differences including a statistical analysis in Supplementary Table 1.

Introduction:

Some examples for English improvement

*** We thank you for this comment and have made the suggested changes and also critically read through our manuscript.

Line 42 “Not only are these foods” change in “These are food not only locally..”

*** The sentence has been rephrased.

Line 48 “on their diverse microbial community.”

*** This has been corrected.

Line 54 you were talking of general properties and then talk specifically of milk, keep staying general and more specific in the successive part.

*** Here, we have added a remark that in general fermentation can remove anti-nutritional factors before we mention the lactose from milk. In this way, we have generalized our statement here.

Line 75 change between in among

Line 83 change between in among

*** These changes have been made.

M &M

Check the spaces between numbers and unit measure and report City and State of the company.

*** We have added the city in Germany where the company is based (for Germany it is uncommon to mention the state or Bundesland).

Line 100-102, why you asked their age and education degree? You should have asked  about the processes (time, temperature, backslopping ..)

*** This has been removed from the manuscript, both in the Methods section and in the Results. Instead, we have added details on the processing procedures that processor use to make the products (Line 106-114).

Line 177-180 these part goes in the results or discussion

*** This has been removed this from the methods section and has been integrated into the discussion (Line 433-436).  

Line 246-249 Do not explain in M&M

*** We rephrased this so that the remarks are linked to the statistical method itself and not directly to the aims of our study or its interpretation. (Line 285-261)

pH was measured but it is not reported in M&M

*** This has been added in the M&M section on sampling (line 119).

Results

Line 253 -257 I don’t think these part about ages and education is useful for the results.

*** We agree with the Reviewer and have removed these results.

Table 1 column Carbohydrates, percentage coefficient of variation is missing

*** This has been added to Table 1.

Line 280 Table 3 is not the table you are referring to and should not be placed in this part of the results

Figure 1 in the caption you don’t have to describe the results (Line 293-302), Line 297: orange and green ellipses not reported.

*** Description of results has been removed from the figure caption and was integrated into the main text; the mention of the elipses had also been removed.

Table 4 you have to introduce table and this is commented later on the manuscript

*** We have expanded on the description of the results that are presented in this table in the main text of the manuscript by rewriting this paragraph (Line 347-355).

Line 311 is not Table 5

Table 5 you should improve this table and comment it, I think you can add the EAR% values of Mabisi and Munkoyo to better understand their nutritional values instead of putting these data on table 1

*** Thank you for this suggestion. The %EAR has been moved to what is now Table 3. Further, text has been added to the Results section to help interpret the Table (Line 282-287).   

Figure 2 why Lactococcus lactic is represented by four colors? I would suggest more descriptions of this figure. Do you have an idea of the species concentration as CFU/g? I think this is of interest since you explain some difference thanks to microbial activity.

*** Since more than one bacterial type was identified as Lactococcus lactis, this species name appears more than once. We have added more description and context for these results, both in the Results section (Line 327-329), as well as in the discussion section (Line 391-392). In this present study, we did not directly measure CFU/g, yet from other work we know that this consistently is around 108 CFU/g. We have added this information to the text (Line 392-393).

Line 338 it is not Table 4 but 3, please adjust the order and place tables in the correct order

*** This has been corrected.

Line 347 I think you mean Table 4, why you chose those nutritional parameters?  And how for example protein and fat contribute to microbial community?

*** We chose the nutritional parameters for which sufficient variation exist among samples to allow for the statistical test, this has been clarified in the text (352-354).

Protein and fat content may influence the microbial community composition since these are resources for microbes to proliferate. Differences in protein/fat content or composition may influence the bacterial community, with different protein/fat level selecting for different microbial groups. This information has been added to the discussion section (Line 400-402).

Table 6 This table has to be moved from this part.

*** This has been corrected.

Discussion

I think that you should provide some information about the composition of maize and milk, in order to explain the Mabisi and Munkoyo differences and consider them as two separated products that you primarily characterize rather than compare. Moreover, we do not know anything about the fermented food process (except for the vessels) that play a role in their compositions.

*** It is indeed very useful to provide information on the raw milk and the unfermented maize porridge to give context to the differences in nutritional composition that we observed between the two product types. Our sampling strategy did not allow for the collection of raw milk or unfermented maize porridge, and we also do not exactly know the quality of the milk nor amounts of maize used by the specific processors. We do now make a general remark that milk is more nutritious than maize, including two references (Line 368-371).

Descriptions on the fermentation process have been added to the methods sections, including the addition of several references to studies that describe the fermentation process of Mabisi and Munkoyo in more detail (Line 106-114).  

Line 379  and 423 you should restate this sentence, some strains are able to produce these vitamins, and which species among those that you found could be able ?

*** this indeed is an important point. We have elaborated on this in the discussion section (Line 441-448). From present work we cannot infer what individual strains are responsible for vitamin production. Bacterial genera are highly variable in how many vitamins they produce. So, a strain that has been identified as belonging to the same genus or family may have very different vitamin production than other strains with the same “name”. Directed and controlled laboratory experiments could provide this insight.

Reviewer 3 Report

The work deal with the microbial and nutritional composition of two Zambian traditional fermented products, one milk-based and one maize-based. For sure the description of the characteristics of the two products could be considered a novelty, and could help in planning of nutritional programs in Zambia, but the paper needs some reorganization and a linguistic check.

Point 1: It’s not correct and useful compare the chemical and nutritional composition of two products based on raw materials completely different. It’s like to compare milk and maize, they are inevitably different, especially in proximate composition, but also regarding minerals and vitamins. So, the Authors should describe separately the two fermented foods, without comparison. The Authors could keep the comparison among the same products from different producers, but a better discussion should be done in this sense.

For these motifs, tables 1 and 2 should be modified and Figure 1 should be eliminated.

Point 2: Information about age, gender, and education level are not useful for the experimentation and should be removed from the results. Instead other important process information (i.e. time and temperature) are lacking for an exhaustive discussion.

Point 3: The Authors said that they found a correlation between levels of contribution to the EAR of various nutrients and variation in microbial community structure. Where did the Authors report these results and demonstrate this statement? Moreover, the Authors attributed the content of B-vitamins (especially in Munkoyo) to the activity of bacteria. The Authors should better discuss this aspect considering the strain dependence of this capability and widely commenting the results of microbial analysis.

Point 4: Other general improvements to the manuscript

Tables 3, 4, 5, 6 are wrong in the order or in the reference in the text

It’s redundant show both table 3 and figure 3

LL 98-99: it’s sufficient to report the number of samples effectively taken, not the planned ones

L 137: “HCl” instead of “HCL”

LL 150-151: check and uniform the “space” between number and unit

LL 267-268: this caption is redundant

L 283: how many grams is one serving of Munkoyo?

LL 352-355: the Authors said that the categorization of nutritional variables explained significant parts of the variation in microbial community composition. But if this statement is true, it’s in contrast with the statement that affirm that the composition of the microbial community is a determinant of the nutritional content of the products (LL 421-424). The Authors should clarify this point.

Point 5: A general linguistic check is needed. Some examples below:

L 35: please reformulate this sentence

L 42: please reformulate this sentence

L 86: “programs” instead of “programmes”

L 91: remove “their”

LL 95-96: please reformulate this sentence

LL 116 231 246 251 307 336 433…..: avoid the use of “we….”

LL 164-165: “according to” instead of “that have been obtained before in”

Author Response

Comments and Suggestions for Authors

The work deal with the microbial and nutritional composition of two Zambian traditional fermented products, one milk-based and one maize-based. For sure the description of the characteristics of the two products could be considered a novelty, and could help in planning of nutritional programs in Zambia, but the paper needs some reorganization and a linguistic check.

*** We thank you for your positive feedback and constructive review that has helped to further improve the manuscript. All suggestions have been incorporated into a revised version of the manuscript. Below, we outline what changes have been made for each of the points you made.

Point 1: It’s not correct and useful compare the chemical and nutritional composition of two products based on raw materials completely different. It’s like to compare milk and maize, they are inevitably different, especially in proximate composition, but also regarding minerals and vitamins. So, the Authors should describe separately the two fermented foods, without comparison. The Authors could keep the comparison among the same products from different producers, but a better discussion should be done in this sense.

For these motifs, tables 1 and 2 should be modified and Figure 1 should be eliminated.

*** The comparison of both product types was mainly meant for the microbial composition. For microbial composition we had previously not found a difference in microbial community structure while we had expected one. This study aims at a more thorough test for this difference. We agree that regarding nutritional composition, differences between the two product types are inevitable given the different raw materials. The Tables describing these differences have been modified by removing the comparison between the product types. After some consideration, we have chosen to keep Figure 1 since it highlights the variation between samples of the same product type (as well as between the two product types), highlighting which nutrients contribute most to this variation. In Figure 1 we did remove the emphasis on the comparison between the two product types and moved the table describing the loadings of PC vectors to the Supplement (Supplementary Table 2). Since some readers may still find this interesting and also since not for all nutrients analysed a difference was found, we do still report on the differences including a statistical analysis in Supplementary Table 1.

Point 2: Information about age, gender, and education level are not useful for the experimentation and should be removed from the results. Instead other important process information (i.e. time and temperature) are lacking for an exhaustive discussion.

*** The information about the processors has been removed. Rather, we have now included a description on the processing methods that are used to produce the products. All processors used a highly similar processing method for Mabisi or Munkoyo. These processing methods have also been described elsewhere and we have included references to this other work. (Line 106-114).

Point 3: The Authors said that they found a correlation between levels of contribution to the EAR of various nutrients and variation in microbial community structure. Where did the Authors report these results and demonstrate this statement? Moreover, the Authors attributed the content of B-vitamins (especially in Munkoyo) to the activity of bacteria. The Authors should better discuss this aspect considering the strain dependence of this capability and widely commenting the results of microbial analysis.

*** We found evidence that differences in microbial community composition could result in differences in nutrient levels in the final product; some microbial communities contribute more B-vitamins through bacterial activity than others. We have rewritten parts of the results section (Line 347-355) to better explain this and also elaborate on this finding in the discussion section (Line 433-436). Further, we have expanded on the discussion of this finding by adding more context. We now mention and cite other studies have also shown that lactic acid bacteria can enhance vitamin content and that the use of fermentation processes can be a strategy to increase nutritional value of raw materials (Line 437-448).

Point 4: Other general improvements to the manuscript

Tables 3, 4, 5, 6 are wrong in the order or in the reference in the text

*** This has been corrected.

It’s redundant show both table 3 and figure 3

*** We have removed the table to the supplementary information.

LL 98-99: it’s sufficient to report the number of samples effectively taken, not the planned ones

*** This has been corrected.

L 137: “HCl” instead of “HCL”

*** This has been corrected.

LL 150-151: check and uniform the “space” between number and unit

*** This has been made uniform.

LL 267-268: this caption is redundant

*** This second copy of the caption has been removed.

L 283: how many grams is one serving of Munkoyo? 183g?

*** Indeed, 183 g. We have added a reference for why we used this amount (Hotz, C., Palaniappan, U., Chileshe, J., Kafwembe, E., & Siamusantu, W. (2011). Nutrition survey in central and eastern provinces, Zambia 2009: Focus on vitamin A and maize intakes, and vitamin A status among women and children. Washington (DC): National Food and Nutrition Commission of Zambia, Lusaka, Zambia.)

LL 352-355: the Authors said that the categorization of nutritional variables explained significant parts of the variation in microbial community composition. But if this statement is true, it’s in contrast with the statement that affirm that the composition of the microbial community is a determinant of the nutritional content of the products (LL 421-424). The Authors should clarify this point.

*** Thank you for pointing this out; we feel this related to your Point 3. We have tried to further clarify this (Line 433-436).

Point 5: A general linguistic check is needed.

*** We have performed a linguistic check of the entire manuscript.

Some examples below:

L 35: please reformulate this sentence

*** This sentence has been reformulated to read ... In many countries locally processed traditional foods exist and these contribute to the diets of their consumers.

L 42: please reformulate this sentence

*** This sentence has been split in multiple sentences.

L 86: “programs” instead of “programmes”

*** This has been corrected.

L 91: remove “their”

*** This has been corrected.

LL 95-96: please reformulate this sentence

*** This sentence has been removed.

LL 116 231 246 251 307 336 433…..: avoid the use of “we….”

*** This has been addressed by reformulating these sentences.

LL 164-165: “according to” instead of “that have been obtained before in”

*** This has been corrected.

Reviewer 4 Report

NUTRIENTS 793426

Nutritional composition and microbial communities of two non-alcoholic traditional fermented beverages from Zambia: A study of Mabisi and Munkoyo

Line 13. Should add: foods “and beverages” since the subject matter is beverages.

Lines 31/32. The keyword “biofortification” does not make sense here. Many NGOs and global alliances of agriculture and nutrition research have interests in working to increase the micronutrient density of staple food crops in poor countries through biofortification. In Africa one needs to evaluate if this is Science or Business, or both! But this leads to long discussions….

Lines 44/76/179/420/425/435: the word “microbes” should be replaced by “microorganisms”.

Lines 63/64. Since previous work by the authors showed that microbial composition may vary with several variations in processing, how was data from each producer in the present trial avoided this bias?

Line 68. It claims Munkoyo is boiled for several hours. Not correct I believe? The mixture maize/roots is placed in boiling water but not boiled for hours. Fermentation lasts 48-72h?

Lines 85-86. Planning nutritional programmes in Zambia; not clear if Zambia has 2 things: 1) Defined Nutritional Requirements of Zambians; 2) Composition of Zambian Foods. Without these 2 instruments it is difficult to plan, since fermented Mabisi and Munkoyo are just complements of a base diet.

Line 87. Numerous studies already identified the nutritional value of microorganisms and also the impact on health, including mental health.

Line 97. Size of samples: 50 mL for Mabisi and 50mg for Munkoyo?

Line 106. “For” and not “food” nutritional analysis.

Line 109. Department of Food Sciences and Nutrition (in Capitals)

Line 110. Water content: delete “content”

Line 115. Incorrect statement “nutritional content”. Either “nutritional value” or better “nutrient value”.

Line 116. By definition “proximate analysis” includes already the nutrients mentioned. Delete these words.

Lines 116-117. Why not mentioned Dry matter content? Surprisingly, Energy levels is missing although mentioned later in Table 1 (with quite large variation mentioned: mean 76, SD 32)!

Line 117. Milk contains small amounts of niacin, pantothenic acid, vitamin B6, vitamin C, and folate although their bioavailability is good. Tested B Vitamins were B1=thiamine, B2=riboflavin, B3=niacin, B6=pyridoxin, and B12= cyanocobalamin. Why B3 and B6 were chosen? Even in industrialized countries, thiamin, riboflavin, vitamin B6, folates and vitamin B12 intakes are frequently lower or in the lower range of recommendations.

Lines 120-122. Not clear. Needs rewriting since the Reference 17 is not the proper one to calculate the amount of 183g. The estimated average requirement (EAR) is the calculated (not measured) amount of a nutrient that is estimated to meet the requirement for a specific criterion of adequacy of half of the healthy individuals of a specific age, sex, and life-stage. But Requirements of Zambian people were not determined, to our knowledge, and nutritional guidelines from Europe or America may not apply, taking into account the nutritional value of African foods. The Dietary Guidelines of Food and Agriculture Organization of the United Nations, 2009, mention Benin, Kenya, Namibia, Nigeria, Seychelles, Sierra Leone, and South Africa but not Zambia.

Line 127. The authors did not take into account that for Milk (Mabisi) the correct and adequate conversion factors, known as N factors, must be 6.38 for milk (Mabisi) and 6.25 for maize (Munkoyo). This may affect conclusions on crude protein evaluations in the present trial. Several authors even use the factor 5.83 for most grains.

Line 173/232. Insert (SD) after standard deviations.

Line176-229. The methodology was previously used by main author (SS), so perhaps could cite his own reference.

Line 233. Student is with capital S.

Line 241/337/341/344/345. (PD) should be after phylogenetic diversity (PD). The word “measure” is redundant since it says indices before.

Line 243. Perhaps write “richness” after Chao1 diversity.

Line 264-266. Surprisingly Mabisi showed lower moisture content than Munkoyo. How is this justified?

Line 269. Table 1. Mabisi showed larger SDs than Munkoyo probably reflecting greater variations, but not explained. pH much lower in Munkoyo. Milk fat varied a lot (3.08-6.98); how come?

Line 283. Table with capitals

Line 287. Suggestion: the order should be better Mabisi on the left and Munkoyo on the right.

Line 289. Figure 1. Orange and green ellipses are not presented on my version.

Line 304. In Table 4. Treatment not Traetment. Why B6 is blank?

Line 306. 2 dots.

Line 322. Reference 24. Is this correct?

Line 324. This Reference is not in the References. EFSA values [33] !! Maize is rich in Zinc, so surprisingly how Mabisi higher than Munkoyo in this mircomineral.

Lines 452/453. “Other recent work”: This should not be on Discussion but somewhere before, with proper Reference.

General Comments

According to the Micronutrient and Food Consumption Survey conducted in two Zambian provinces, 64% and 79% of households consumed adequate dietary quality and quantity, which is considered quite reasonable. Nutrients 2019, 11(2), 288; https://doi.org/10.3390/nu11020288

The role of fermented milks and foods in human nutrition is well documented. The best way of preventing micronutrient malnutrition is to ensure consumption of a balanced diet that is adequate in every nutrient. Promotion of diet diversity is necessary. Unfortunately, this is far from being achievable everywhere since it requires universal access to adequate food and appropriate dietary habits.

Fermentation process is initiated by unplanned indigenous microorganisms or through ‘backslopping’ with mixed or unknown cultures. Often, this type of fermentation may be slow or unpredictable. The research included age of processors and other details but not clear the cultivation/fermentation time of the 2 products investigated.

Nothing mentioned on the origin of milk. Cow’s milk?

Even raw milk from “certified,” “organic,” or “local” dairies is not guaranteed to be safe. The presence of germs in raw milk is unpredictable. Raw milk can carry dangerous germs, such as Brucella, Campylobacter, Cryptosporidium, E. coli, S. aureus, Listeria monocytogenes, Mycobacterium bovis, and Salmonella, which can pose serious health risks. Since there is no alcohol production were these supposed to be inactivated by lactic acid and antimicrobial peptides (bacteriocins) without evaluation?

Was alcohol (ethanol) measured since it is expected to have residual values?  Acetaldehyde (ACH- Group 1 carcinogen for humans) may also be produced during the milk fermentation process and their carcinogenic potential contributes to the high incidence of oesophageal cancer. Is this condition known in the Zambian regions investigated?

The differences between fermented milks are based on the location of tribal communities and the different production processes. Yeast and lactic acid bacteria (LAB) have been reported to be the dominant microbial species in African fermented milk. The most commonly isolated yeasts are Candida and Saccharomyces spp. Besides bacteria were yeasts, mould, or fungi researched?

Microorganisms and their enzymes are able to some extent and with varied efficiency to degrade mycotoxins to less- or non-toxic products. Mycotoxins were not tested so safety of fermented Mabisi and Munkoyo cannot be 100% guaranteed.

Present results showed that in the samples there were two to four dominant bacterial species: Lactobacillus, Lactococcus, Streptococcus, Enterobacter, Klebsiella, or Acetobacter. Were commonly other bacteria such as Leuconostoc and Enterococcus spp not found?

Microbial community structure in these products is neither a simple consequence of the raw materials used (milk or maize), nor the particular suite of microbes available in the environment but that anthropogenic variables (e.g., competition among sellers or organoleptic preferences by different processors) are important in shaping the microbial community structures.

In what refers to the supply of nutrients to meet requirements, the used EAR for adults is an estimate of maintenance needs. Like the probability approach, the EAR cut-point method requires knowledge of the median requirement (the EAR) for the nutrient and the distribution of usual intakes in the population. Individuals in a group vary both in the average amounts of a nutrient they consume and in their requirements for the nutrient. In few words, the EAR method described for many years is a good attempt to diagnose requirements but does not substitute other studies. Reliable and valid methods of food composition analysis are crucial in determining the intake of a nutrient needed to meet a requirement. For several B vitamins and choline, analytic methods to determine the content of the nutrient in food have serious limitations. The most valid intake data are those collected from the metabolic study protocols.

Prevalence of adequate intakes were determined using the estimated average requirement (EAR) cut-point method. For some nutrients (e.g. calcium, vitamin D, pantothenic acid, biotin, and choline), the amount and quality of data currently available for both nutrient intakes and requirements may not be sufficient to apply these statistical models in their entirety for purposes of research and policy. Moreover, in addition to assessing nutrient intakes, assessment of health and nutritional status of groups or individuals must include biochemical, clinical, and anthropometric data.

Food intake is generally studied in terms of adequacy of nutrients. However, foods contain other chemical compounds, some of which are established, some are poorly characterized, and others being completely unknown cannot be measured. Food composition tables of many African foods and their accuracy is described as “conflicting”. Food composition data are not even routinely reported. Therefore stating percentage of a nutrient intake in relation to total intake may not be adequate or correct.

On the present research factor analysis was used for exploring the existence of consumption pattern of food and nutrients and their relationship with the nutritional status of Zambian rural adult population.

Author Response

NUTRIENTS 793426

Nutritional composition and microbial communities of two non-alcoholic traditional fermented beverages from Zambia: A study of Mabisi and Munkoyo

*** We thank the reviewer for this detailed and very constructive and inspirational review. It has allowed us to further improve the manuscript and to expand its scope. We have addressed all comments, below is an itemized response to each comment made.

Line 13. Should add: foods “and beverages” since the subject matter is beverages.

*** This has been added.

Lines 31/32. The keyword “biofortification” does not make sense here. Many NGOs and global alliances of agriculture and nutrition research have interests in working to increase the micronutrient density of staple food crops in poor countries through biofortification. In Africa one needs to evaluate if this is Science or Business, or both! But this leads to long discussions….

*** We agree with the reviewer and have removed this key word. We meant in-situ fortification, which is a term used in some of the food science literature. But yes, we agree that this could lead to long discussion that are beyond the scope of our work.

Lines 44/76/179/420/425/435: the word “microbes” should be replaced by “microorganisms”.

*** This has been corrected.

Lines 63/64. Since previous work by the authors showed that microbial composition may vary with several variations in processing, how was data from each producer in the present trial avoided this bias?

*** We have now selected processors that use (roughly) the same processing method. This has been highlighted in the methods section (Line 106-114) and we reflect on this in the discussion (Line 417-419).

Line 68. It claims Munkoyo is boiled for several hours. Not correct I believe? The mixture maize/roots is placed in boiling water but not boiled for hours. Fermentation lasts 48-72h?

*** Maize is mixed with water and this mixture is indeed boiled for several hours (on average 2 hours), to allow for gelatinization of starch. After cooling, the roots are added and after this, fermentation takes place. This has now been explained in the methods section (Line 106-114).

Lines 85-86. Planning nutritional programmes in Zambia; not clear if Zambia has 2 things: 1) Defined Nutritional Requirements of Zambians; 2) Composition of Zambian Foods. Without these 2 instruments it is difficult to plan, since fermented Mabisi and Munkoyo are just complements of a base diet.

*** We have changed the text and deleted nutrition programmes and revered to the development of food based dietary guidelines currently ongoing in Zambia, and that these 2 fermented beverages could play a role in these (Line 93-94).

Line 87. Numerous studies already identified the nutritional value of microorganisms and also the impact on health, including mental health.

*** Yes, you are right. We have removed this sentence.

Line 97. Size of samples: 50 mL for Mabisi and 50mg for Munkoyo?

*** Indeed, for both Munkoyo and Mabisi we collected 50ml.

Line 106. “For” and not “food” nutritional analysis.

*** This has been corrected.

Line 109. Department of Food Sciences and Nutrition (in Capitals)

*** This has been corrected.

Line 110. Water content: delete “content”

*** Done.

Line 115. Incorrect statement “nutritional content”. Either “nutritional value” or better “nutrient value”.

*** We have changed to nutrient value.

Line 116. By definition “proximate analysis” includes already the nutrients mentioned. Delete these words.

*** These words have been deleted.

Lines 116-117. Why not mentioned Dry matter content? Surprisingly, Energy levels is missing although mentioned later in Table 1 (with quite large variation mentioned: mean 76, SD 32)!

*** Dry matter content has been added to Table 1. Indeed we found quite a variation in energy levels. Perhaps the reason for this could be that the milk quality of the raw milk used varies considerably between the different processors, especially those of sample 1 and sample 8, since these seem outliers.  

Line 117. Milk contains small amounts of niacin, pantothenic acid, vitamin B6, vitamin C, and folate although their bioavailability is good. Tested B Vitamins were B1=thiamine, B2=riboflavin, B3=niacin, B6=pyridoxin, and B12= cyanocobalamin. Why B3 and B6 were chosen? Even in industrialized countries, thiamin, riboflavin, vitamin B6, folates and vitamin B12 intakes are frequently lower or in the lower range of recommendations.

*** Vitamin B3 and B6 were chosen since previous pilot work suggested that especially levels of Vitamin B3 and Vitamin B6 could be increased through microbial activity.

Lines 120-122. Not clear. Needs rewriting since the Reference 17 is not the proper one to calculate the amount of 183g. The estimated average requirement (EAR) is the calculated (not measured) amount of a nutrient that is estimated to meet the requirement for a specific criterion of adequacy of half of the healthy individuals of a specific age, sex, and life-stage. But Requirements of Zambian people were not determined, to our knowledge, and nutritional guidelines from Europe or America may not apply, taking into account the nutritional value of African foods. The Dietary Guidelines of Food and Agriculture Organization of the United Nations, 2009, mention Benin, Kenya, Namibia, Nigeria, Seychelles, Sierra Leone, and South Africa but not Zambia.

**We agree with the reviewer that these sentences were not clear and have changed these accordingly. The 183g was the average portion size found in a study conducted in Zambia several years ago (Line 133). We clarified this and added the reference (Hotz, C., Palaniappan, U., Chileshe, J., Kafwembe, E., & Siamusantu, W. (2011). Nutrition survey in central and eastern provinces, Zambia 2009: Focus on vitamin A and maize intakes, and vitamin A status among women and children. Washington (DC): National Food and Nutrition Commission of Zambia, Lusaka, Zambia.)

Line 127. The authors did not take into account that for Milk (Mabisi) the correct and adequate conversion factors, known as N factors, must be 6.38 for milk (Mabisi) and 6.25 for maize (Munkoyo). This may affect conclusions on crude protein evaluations in the present trial. Several authors even use the factor 5.83 for most grains.

** We actually used 6.38 conversion factor for Mabisi and 6.25 for Munkoyo. This has been corrected in the manuscript methods section (Line 143-144).

Line 173/232. Insert (SD) after standard deviations.

*** This has been added.

Line176-229. The methodology was previously used by main author (SS), so perhaps could cite his own reference.

*** Thank you for the suggestion. We felt that it is good to be able to get an idea of the methods without having to retrieve another publication. Should the editor feel that the manuscript is too long, we could still remove this part.

Line 233. Student is with capital S.

*** This has been corrected.

Line 241/337/341/344/345. (PD) should be after phylogenetic diversity (PD). The word “measure” is redundant since it says indices before.

*** Corrected

Line 243. Perhaps write “richness” after Chao1 diversity.

*** Corrected

Line 264-266. Surprisingly Mabisi showed lower moisture content than Munkoyo. How is this justified?

*** This could have to do with the fact that Mabisi contains more fat and protein (casein) than Munkoyo. Further, maize is mixed with water at the start of processing. The amount of water used may vary and affect the moisture content of the final Munkoyo product.

Line 269. Table 1. Mabisi showed larger SDs than Munkoyo probably reflecting greater variations, but not

explained. pH much lower in Munkoyo. Milk fat varied a lot (3.08-6.98); how come?

*** We do not have a good explanation for this. The quality and content of the raw milk may vary between processors since they are using the milk from their own cows. The quality of the maize used may be more uniform between processors.

Line 283. Table with capitals

*** This has been corrected.

Line 287. Suggestion: the order should be better Mabisi on the left and Munkoyo on the right.

*** We modified the table as suggested and moved this table to the Supplement as per suggestion of another Reviewer (Supplementary Table 3).

Line 289. Figure 1. Orange and green ellipses are not presented on my version.

*** We have removed the mention of the ellipses from the legend.

Line 304. In Table 4. Treatment not Traetment. Why B6 is blank?

*** B6 is blank since for this variable insufficient variation exists to perform the statistical test. We have now removed B6 from the table and explained why some variables have not been included in this particular analysis (Line XXX)

Line 306. 2 dots.

*** One dot has been removed.

Line 322. Reference 24. Is this correct?

*** No, this reference is not correct. Should be the reference to the WHO guidelines of food fortification (2006).

Line 324. This Reference is not in the References. EFSA values [33]!! Maize is rich in Zinc, so surprisingly how Mabisi higher than Munkoyo in this mircomineral.

*** The correct reference has been added to the footer of this Table (now Table 3). European Food Safety, A. Dietary Reference Values for nutrients Summary report. EFSA Supporting Publications 2017, 14, e15121E, doi:10.2903/sp.efsa.2017.e15121.

Indeed maize and therefor Munkoyo is rich in Zinc. Apparently, the raw milk used here was also high in Zinc.

Lines 452/453. “Other recent work”: This should not be on Discussion but somewhere before, with proper Reference.

*** We have now also mentioned this aspect in the Introduction (Line XXX-XXX) and we have added references. Futher, we have expanded on this point in the discussion section (Line XXX-XXX).  

General Comments

According to the Micronutrient and Food Consumption Survey conducted in two Zambian provinces, 64% and 79% of households consumed adequate dietary quality and quantity, which is considered quite reasonable. Nutrients 2019, 11(2), 288; https://doi.org/10.3390/nu11020288

*** Indeed for most micronutrients reasonable levels of micronutrient intakes were found in this study. On the other hand, the study identified inadequate intakes for especially Vitamin B3, Vitamin B12 and Iron. These B-vitamins are of interest to our study since microbial activity could increase their levels in the final products. We have added text to the Discussion section (Line 384-387).

The role of fermented milks and foods in human nutrition is well documented. The best way of preventing micronutrient malnutrition is to ensure consumption of a balanced diet that is adequate in every nutrient. Promotion of diet diversity is necessary. Unfortunately, this is far from being achievable everywhere since it requires universal access to adequate food and appropriate dietary habits.

*** We agree completely here.

Fermentation process is initiated by unplanned indigenous microorganisms or through ‘backslopping’ with mixed or unknown cultures. Often, this type of fermentation may be slow or unpredictable. The research included age of processors and other details but not clear the cultivation/fermentation time of the 2 products investigated.

*** We agree that important information on the processing is missing. We have now added this information to the methods section (Line 106-114) and have added several references to studies that have investigated the fermentation process of the two products.

Nothing mentioned on the origin of milk. Cow’s milk?

*** Indeed, cows milk. We have added this in the text (Line 62 and Line 107).

Even raw milk from “certified,” “organic,” or “local” dairies is not guaranteed to be safe. The presence of germs in raw milk is unpredictable. Raw milk can carry dangerous germs, such as Brucella, Campylobacter, Cryptosporidium, E. coli, S. aureus, Listeria monocytogenes, Mycobacterium bovis, and Salmonella, which can pose serious health risks. Since there is no alcohol production were these supposed to be inactivated by lactic acid and antimicrobial peptides (bacteriocins) without evaluation?

*** Low pH generally makes a product safe against pathogen proliferation and this is also true for these products. We have added an explanation on this in the discussion section (Line 396-398). In other (unpublished) work we found that pathogenic bacteria are unable to invade the final products even when spiked at high numbers. Also, after adding pathogenic bacteria at the start of fermentation, these do not survive into the final product. We are still investigating the mechanisms by which this happens and what conditions DO allow invasion of pathogenic bacteria – knowing which processing and storage conditions to avoid.

Was alcohol (ethanol) measured since it is expected to have residual values?  Acetaldehyde (ACH- Group 1 carcinogen for humans) may also be produced during the milk fermentation process and their carcinogenic potential contributes to the high incidence of oesophageal cancer. Is this condition known in the Zambian regions investigated?

** We did not measure the ethanol levels and the oesophageal cancer incidence is unknown in Zambia. This maybe a possible question to be addressed in future studies. More generally, other work focussing on aroma profiles of Munkoyo and Mabisi have shown that the levels of alcohol are relatively low, in the order of 0.5% and under.

The differences between fermented milks are based on the location of tribal communities and the different production processes. Yeast and lactic acid bacteria (LAB) have been reported to be the dominant microbial species in African fermented milk. The most commonly isolated yeasts are Candida and Saccharomyces spp. Besides bacteria were yeasts, mould, or fungi researched?

*** Previous work on the microbiology of fermentation of these two products did include the analysis of yeast and moulds. In this work yeast was not found in most of the samples. We therefor decided not to include this analysis here. Further, our interest is in the potential contribution to B-vitamin levels by fermentation – we feel that the most dominant micro-organisms (the bacteria) are of most interest. We do mention now in the manuscript that we did not include the analysis of yeast.

Microorganisms and their enzymes are able to some extent and with varied efficiency to degrade mycotoxins to less- or non-toxic products. Mycotoxins were not tested so safety of ermented Mabisi and Munkoyo cannot be 100% guaranteed.

*** Indeed, maize can be contaminated with mycotoxins and there is some evidence that fermentation with lactic acid bacteria may reduce mycotoxin levels. On the other hand, mycotoxins are likely not fully eliminated and without proper testing the safety with regards to mycotoxins may not be guaranteed. We have now mentioned mycotoxins in the discussion section (line 445-447).

Present results showed that in the samples there were two to four dominant bacterial species: Lactobacillus, Lactococcus, Streptococcus, Enterobacter, Klebsiella, or Acetobacter. Were commonly other bacteria such as Leuconostoc and Enterococcus spp not found?

*** This is an interesting question. In other work on the same products, but sampled in other parts of Zambia, we did find these bacterial species, although not at very high abundance. Apparently the local environment of Mkushi where we did our sampling did not include this bacterial flora.

Microbial community structure in these products is neither a simple consequence of the raw materials used (milk or maize), nor the particular suite of microbes available in the environment but that anthropogenic variables (e.g., competition among sellers or organoleptic preferences by different processors) are important in shaping the microbial community structures.

*** We fully agree, processing practice likely is a very important selection pressure shaping the microbial community composition. Other work of our group that is now under review shows evidence for this.

In what refers to the supply of nutrients to meet requirements, the used EAR for adults is an estimate of maintenance needs. Like the probability approach, the EAR cut-point method requires knowledge of the median requirement (the EAR) for the nutrient and the distribution of usual intakes in the population. Individuals in a group vary both in the average amounts of a nutrient they consume and in their requirements for the nutrient. In few words, the EAR method described for many years is a good attempt to diagnose requirements but does not substitute other studies. Reliable and valid methods of food composition analysis are crucial in determining the intake of a nutrient needed to meet a requirement. For several B vitamins and choline, analytic methods to determine the content of the nutrient in food have serious limitations. The most valid intake data are those collected from the metabolic study protocols.

Prevalence of adequate intakes were determined using the estimated average requirement (EAR) cut-point method. For some nutrients (e.g. calcium, vitamin D, pantothenic acid, biotin, and choline), the amount and quality of data currently available for both nutrient intakes and requirements may not be sufficient to apply these statistical models in their entirety for purposes of research and policy. Moreover, in addition to assessing nutrient intakes, assessment of health and nutritional status of groups or individuals must include biochemical, clinical, and anthropometric data.

*** This is acknowledged but these suggested aspects are out of the scope of this study. We only use % of the EAR as a tool to see what kind of contribution the beverages could have. We agree with you that more research on this is needed, especially in low and middle income countries. Obtaining funding for these kind of studies remains difficult, let alone the possibility to conduct these studies under the same rigour as in high income countries (for metabolic studies for example).

Food intake is generally studied in terms of adequacy of nutrients. However, foods contain other chemical compounds, some of which are established, some are poorly characterized, and others being completely unknown cannot be measured. Food composition tables of many African foods and their accuracy is described as “conflicting”. Food composition data are not even routinely reported. Therefore stating percentage of a nutrient intake in relation to total intake may not be adequate or correct.

On the present research factor analysis was used for exploring the existence of consumption pattern of food and nutrients and their relationship with the nutritional status of Zambian rural adult population.

***This is acknowledged, however in this manuscript, factor analysis was only used to determine variation in nutritional components between the products and not to explore the contribution of the nutrients towards achieving EAR and relationship with nutritional status.

Round 2

Reviewer 2 Report

The manuscript was improved according to the suggestions, results are clearer, tables and figures were modified providing a better description of the results. Sentences were added in order to better explains and justify the results. However, in my opinion it is not very clear if the microorganisms are responsible for vitamins production or vitamins affect microbial community.

Still some points to be addressed:

Check the spaces after the unit system

e.g line 133 183 g, line 138 60kg

line 167 40 mL 4mL

Abstract

Line 14: delete “including “

Introduction:

Line 40: please invert “are there” in there are food-based dietary

Line 44-45: Move further at the beginning of the sentence

Line 50: add commas before and after thus

Line 63-64: add commas.

Line 64: change in previous works

M&M

Line 133-134: Maybe “for” instead of towards?

Line 252: yeasts instead of yeast

Results

Table 1 and 2 are poorly described, some comments about the variability of products of the same fermented foods could be added, and the pH commented, since it is an important parameters for fermented food.

Line 310: “differences between the two products as indicated with small blue circles” change in “small blue circles distribution.”

Line 388  change in “ this may be applied”

Line 393: verb is missing

Line 413: cite the work soon after you mention it

Line 418: change in focusing, without double s

Line 423: add comma after back-sloppings

Line 437: change in “would be applied”

Line 438: Change “added” with “increased”

Line 438 : Lactococcus lactis in italic

Line 467: Change in “However, we ..”  and “to especially determine the nutritional and microbial aspects in the fermentation process”

Author Response

The manuscript was improved according to the suggestions, results are clearer, tables and figures were modified providing a better description of the results. Sentences were added in order to better explains and justify the results. However, in my opinion it is not very clear if the microorganisms are responsible for vitamins production or vitamins affect microbial community.

*** We thank the Reviewer for the positive feedback and the interest in our work.

We agree that our results are inconclusive with respect to showing whether microbial community composition affects vitamin levels in the final product or the other way around. Other work suggests that lactic acid bacteria increase levels of B-vitamins in fermented foods by adding these vitamins to the raw materials. Based on that, we speculate that indeed microbial community composition affects vitamin levels in the final product. Elucidating this would require controlled (laboratory) experiments. In the discussion we now highlight this more clearly and have added text (Line 452-462, also see Line 462-470).

Still some points to be addressed:

Check the spaces after the unit system

e.g line 133 183 g, line 138 60kg

line 167 40 mL 4mL

*** Throughout, we have removed spaces between the number and their unit, unless the unit was written out in full.

Abstract

Line 14: delete “including “

*** The sentence has been reformulated.

Introduction:

Line 40: please invert “are there” in there are food-based dietary

*** These words have been remove since they are redundant.

Line 44-45: Move further at the beginning of the sentence

*** Further is now at the beginning of the sentence.

Line 50: add commas before and after thus

*** The sentence has been reformulated.

Line 63-64: add commas.

*** A comma was added after hours.

Line 64: change in previous works

*** We changed to previous research.

M&M

Line 133-134: Maybe “for” instead of towards?

*** This has been changes in ... to reach...

Line 252: yeasts instead of yeast

*** This has been changed.

Results

Table 1 and 2 are poorly described, some comments about the variability of products of the same fermented foods could be added, and the pH commented, since it is an important parameters for fermented food.

*** We agree with the Reviewer and have added text to the Results section (Line 275-279) as well as to the Discussion section (Line 374-379). The variability between samples of the same product type is likely driven by the variability of raw materials used by processors (a factor we did not control for). Among the Mabisi samples, two remarkable outliers have lower moisture content than the other samples, leading to high values of other proximate composition parameters and lower Vitamin B2, Vitamin B3 and Calcium content. While processors have affirmed they used the Tonga-type method, these two processors may have used another method for Mabisi processing, during which whey is removed thus lowering the moisture content.

The pH for Mabisi is one pH unit higher than for Munkoyo, which is in line with previous findings. Generally, a pH below 4.5 is considered safe as it supresses growth of (most) pathogenic bacteria. The fact that for one Mabisi sample a pH of 4.6 was observed shows that in particular for Mabisi the pH needs to be monitored to ensure the safety of its consumption. We have added text to the discussion section (Line 407-411).

Line 310: “differences between the two products as indicated with small blue circles” change in “small blue circles distribution.”

*** These words have been deleted.

Line 388  change in “ this may be applied”

*** We changed to .... This positive impact may apply ....

Line 393: verb is missing

*** “are” has been added.

Line 413: cite the work soon after you mention it

*** A citation has been added in line 413.

Line 418: change in focusing, without double s

*** We changed to US English spelling.

Line 423: add comma after back-sloppings

*** A comma has been added.

Line 437: change in “would be applied”

*** This sentence has been rephrased. 

Line 438: Change “added” with “increased”

*** This sentence has been rephrased. 

Line 438 : Lactococcus lactis in italic

*** This has been changed.

Line 467: Change in “However, we ..”  and “to especially determine the nutritional and microbial aspects in the fermentation process”

*** This sentence was rephrased: We recommend more research to include determination of the nutritional composition of raw materials and end-products of fermentation to quantify the addition of nutrients by fermenting microbes and to conduct more genomic analysis for B vitamin production.

Reviewer 3 Report

The work was improved according to the suggestions of the reviewers, in particular regarding a reorganization of the work, including tables and figures. The manuscript is now clearer, anyway some other improvements could be done, including a further english check.

Point 1: Since the Authors have better explained the correlation between processing and nutritional variables and variation in microbial community, in my opinion it’s not possible to affirm that the final concentration of B-vitamins contributes to microbial variation. Also because the Authors affirm that the same vitamins are produced by bacteria.

Point 2: The Authors should better discuss the data variability among the same products, maybe adding the variation coefficient of the means showed in tables 1 and 2.

Point 3: Other general improvements to the manuscript

Check and uniform the “space” between number and unit (i.e. LL 133 and 145, L 149, L 167, L 176, L 205 etc).

LL 180-181: add the reference after “according to” and before “[25]”.

Table 3: check and uniform the significant digits.

L 411: “(Chao1 ranged from 166 to 640) [43]” instead of “(Chao1 ranged from 166 to 640, [43])”.

L 438: “Lactococcus lactis” instead of “Lactococcus lactis”

Point 4: A general linguistic check is needed again. Some examples below:

LL 101-102: please reformulate this sentence.

L 133: please reformulate this sentence.

LL 231-232: please reformulate this sentence.

L 393: verb absent.

L 395: “was” instead of “were”.

Author Response

The work was improved according to the suggestions of the reviewers, in particular regarding a reorganization of the work, including tables and figures. The manuscript is now clearer, anyway some other improvements could be done, including a further english check.

*** We thank the Reviewer for the constructive comments that show high interest in our work. English language had been checked and we have addressed the other points by revising the manuscript. Below, we respond to each comment made.

Point 1: Since the Authors have better explained the correlation between processing and nutritional variables and variation in microbial community, in my opinion it’s not possible to affirm that the final concentration of B-vitamins contributes to microbial variation. Also because the Authors affirm that the same vitamins are produced by bacteria.

*** We thank the Reviewer for pointing out that this is not clear. Indeed, our results do not allow to distinguish whether microbial community composition drives final vitamin levels or that vitamin levels drive microbial community composition. However, the fact that lactic acid bacteria have been found to add vitamins during fermentation, suggests that microbial community composition may affect this increase. In this way, our results warrant future work (controlled experiments) to investigate this further. We have elaborated on this in the Discussion section (Line 452-462, also see Line 462-470).  

Point 2: The Authors should better discuss the data variability among the same products, maybe adding the variation coefficient of the means showed in tables 1 and 2.

*** We agree with the Reviewer and have added text to the Results section as well as to the Discussion section. The variability between samples of the same product type is likely driven by the variability of raw materials used by processors (a factor we did not control for). Among the Mabisi samples, two remarkable outliers have lower moisture content than the other samples, leading to high values of other proximate composition parameters and lower Vitamin B2, Vitamin B3 and Calcium content. While processors have affirmed they used the Tonga-type method, these two processors may have used another method for Mabisi processing, during which whey is removed thus lowering the moisture content. We have added the %CV in Tables 1 and 2, a description of these results in the Results section (Line 275-279) and elaborated on this in the Discussion section (Line 374-379).

Point 3: Other general improvements to the manuscript

Check and uniform the “space” between number and unit (i.e. LL 133 and 145, L 149, L 167, L 176, L 205 etc).

*** Throughout, we have removed spaces between the number and their unit, unless the unit was written out in full.

LL 180-181: add the reference after “according to” and before “[25]”.

*** Moreno et al. has been added before [25].

Table 3: check and uniform the significant digits.

*** This has been checked and corrected.

L 411: “(Chao1 ranged from 166 to 640) [43]” instead of “(Chao1 ranged from 166 to 640, [43])”.

*** This has been corrected.

L 438: “Lactococcus lactis” instead of “Lactococcus lactis”

*** This species name is now in italics.

Point 4: A general linguistic check is needed again. Some examples below:

LL 101-102: please reformulate this sentence.

*** The sentence has been reformulated.

L 133: please reformulate this sentence.

*** This sentence has been reformulated.

LL 231-232: please reformulate this sentence.

**** This sentence has been reformulated

L 393: verb absent.

*** “are” has been added.

L 395: “was” instead of “were”.

*** the sentence has been changed in ... while the microbial community composition in Munkoyo samples was more abundant in...